# Nasal bots carry relevant titers of CWD prions in naturally infected white-tailed deer

Paulina Soto[1,2], Francisca Bravo-Risi[1,2], Carlos Kramm [1], Nazaret Gamez[1], Rebeca Benavente [1], Denise L Bonilla [3], J Hunter Reed [4], Mitch Lockwood[4], Terry R Spraker[5], Tracy Nichols[3] & Rodrigo Morales [1,2]✉

## Abstract

**Chronic wasting disease (CWD) is a prion disease affecting farmed and free-ranging cervids. CWD is rapidly expanding across North America and its mechanisms of transmission are not completely understood. Considering that cervids are commonly afflicted by nasal bot flies, we tested the potential of these parasites to transmit CWD. Parasites collected from naturally infected white-tailed deer were evaluated for their prion content using the protein misfolding cyclic amplification (PMCA) technology and bioassays. Here, we describe PMCA seeding activity in nasal bot larvae collected from naturally infected, nonclinical deer. These parasites efficiently infect CWD-susceptible mice in ways suggestive of high infectivity titers. To further mimic environmental transmission, bot larvae homogenates were mixed with soils, and plants were grown on them. We show that both soils and plants exposed to CWD-infected bot homogenates displayed seeding activity by PMCA. This is the first report describing prion infectivity in a naturally occurring deer parasite. Our data also demonstrate that CWD prions contained in nasal bots interact with environmental components and may be relevant for disease transmission.**

**Keywords** Chronic Wasting Disease; Nasal Bots; Disease Vectors; White-tailed Deer; Prions
**Subject Categories** Evolution & Ecology; Microbiology, Virology & Host Pathogen Interaction

## Introduction

Prion diseases, also known as transmissible spongiform encephalopathies (TSEs), are fatal neurodegenerative disorders affecting mammals (Prusiner, 1998; Collinge, 2001). Although rare in humans, prion diseases can manifest in epidemic proportions in captive and wild animals (Collinge, 2001). Chronic wasting disease (CWD) exemplifies the problem that prion diseases can cause to animal populations and associated industries. CWD was first identified in Colorado in 1967 and has uncontrollably expanded across North America. Currently, 31 states from the United States and four Canadian provinces have reported cases of CWD in captive or wild cervid populations, or both (https://www.usgs.gov/media/images/distribution-chronic-wasting-disease-north-america-0). Unfortunately, this prion disease is not restricted to a single animal species. In fact, multiple cervid species have been described to be susceptible to CWD prions in either natural or experimental conditions (Haley and Hoover, 2015). More recently, CWD has been identified in wild Scandinavian reindeer, moose, and red deer populations (Benestad et al, 2016; Vikøren et al, 2019). North American and European CWD prions display different properties, suggesting that this infectious agent may have spontaneously and independently aroused on each continent generating new strains (Nonno et al, 2020; Benestad and Telling, 2018; Bian et al, 2021).

Despite decades of research, several questions regarding CWD remain unanswered, especially in terms of the natural transmission and dissemination of infectious prions. Several hypotheses have been formulated to explain CWD transmission, including horizontal and vertical transmission mechanisms (Nalls et al, 2013; Morales et al, 2013; Bravo-Risi et al, 2021; Nalls et al, 2021) as well as environmental contamination (Saunders et al, 2012; Bartelt-Hunt and Bartz, 2013; Escobar et al, 2020; Zabel and Ortega, 2017). The latter is supported by extensive research and plays a critical role in CWD epidemiology. Compelling evidence suggests that CWD infectivity is most likely deposited in the environment by the binding of prions to a wide array of surfaces, including soil, plants, and farm implements, among others (Zabel and Ortega, 2017; Miller et al, 2004; Mathiason et al, 2009; Bartelt-Hunt and Bartz, 2013; Gough et al, 2014; David Walter et al, 2011; Miller and Williams, 2003). Unfortunately, the specific contribution each environmental component plays as a disease vector is not yet clear. Currently identified and hypothesized fomites involved in CWD transmission include soil, plants, water and other natural and human made elements shown to experimentally and naturally bind or contain infectious prions (Nichols et al, 2009; Plummer et al, 2018;

[1]Department of Neurology, The University of Texas Health Science Center at Houston, Houston, TX, USA. [2]Centro Integrativo de Biologia y Quimica Aplicada (CIBQA), Universidad Bernardo O'Higgins, Santiago, Chile. [3]United States Department of Agriculture, Animal Plant Health Inspection Service, Veterinary Services, Fort Collins, CO, USA. [4]Texas Parks and Wildlife Department, Kerrville, TX, USA. [5]Colorado State University Diagnostic Medical Center, College of Veterinary Medicine and Biomedical Sciences, Colorado State University, Fort Collins, CO, USA. ✉E-mail: Rodrigo.MoralesLoyola@uth.tmc.edu

Pritzkow et al, 2018). In addition, other living entities such as insects (Wisniewski et al, 1996; Haigh et al, 2002; Haley et al, 2021), annelids (Pritzkow et al, 2021), predators (Escobar et al, 2020; Nichols et al, 2015) and scavengers (VerCauteren et al, 2012) have also been hypothesized to either participate as vectors of CWD or broaden the host range of prions. Relevant to the latter, the role of parasites as potential vectors for CWD transmission has been understudied.

Previous reports show that flies (*Sacrophaga carnaria*) fed with brains from prion-infected hamsters can transmit disease when experimentally administered to naive animals (Post et al, 1999). Also, evidence of disease-associated prions has been found in mites collected from hay in farms housing scrapie-infected sheep (Wisniewski et al, 1996). The role of ticks as CWD vectors has been proposed, albeit with contradictory results. Shikiya et al. reported that ticks (*D. andersoni*) experimentally exposed to prion-infected hamsters were inefficient in transmitting prion disease (Shikiya et al, 2020). Ticks collected in this experiment lacked seeding activity as evaluated through the Protein Misfolding Cyclic Amplification (PMCA) technique. In comparison, another report described the presence of real-time quaking-induced conversion (RT-QuIC) seeding activity in winter ticks (*Dermacentor albipictus*) collected from naturally affected North American elk (Haley et al, 2021). More recently, prion seeding activity has been found in lab fed and naturally occurring black-legged ticks (*Ixodes scapularis*) (Inzalaco et al, 2023). Regardless of these positive findings in relevant samples, the specific infectivity titers contained in these parasites is unknown. Along this line, the potential of ticks to transmit disease needs to be carefully evaluated considering the low-infectivity titers in blood (Elder et al, 2013), and the way ticks interact with naive animals (Mooring and Samuel, 1998; Hirth, 1977). All cervids are afflicted by a wide variety of parasites including fleas, ticks, nasal bots, and several species of lung, stomach, muscle, liver and arterial worms (Haley et al, 2021; Haigh et al, 2002; Paddock and Yabsley, 2007; Thompson, 2001; Filip and Demiaszkiewicz, 2016; Akritidis, 2011). The load of prion infectivity in deer parasites, and the relationship between parasitic infection and CWD incidence has not been reported.

Flies of the subfamily Oestrinae (nose bot flies) are common parasitic species afflicting mammals (Cogley and Anderson, 1981). They have unique life cycles where their eggs develop and hatch internally in the female and then first instar larvae are ejected within a liquid into the host. The larvae then develop into second and third instars that migrate through the host respiratory system before dropping from the host to pupate on the ground (Appendix Fig. S1 and Morrondo et al, 2021). This process can be stressful for the host. When under attack, hosts may stop grazing and try to hide their muzzles to stop the flies from larvipositing. Moreover, hosts may have complications from numerous larval bots in their system and occasionally the mature larvae may get lodged in the host creating secondary infection and possible death (Mullen and Durden, 2019). The genus Cephenemyia (deer nose bots) are bot flies that specifically parasitize cervids. There are eight species in the genus and five species in North America, including *C. apicata*, *C. jellisoni*, *C. pratti*, *C. phobifer*, and *C. trompe*. *C. phobifer* (*C. phobifera*) is the common species in the eastern part of North America while the others populate the west. *C. phobifer* parasitize white-tailed deer, mule deer, and moose (Johnson et al, 1983). Cephenemyia species are most often found as larvae in clusters in the host retropharyngeal pouches in the throat near posterior aspects of the soft palate (Bennet and Sabrosky, 1962). Unlike as previously thought for the entire subfamily Oestrinae, at least two species, *C. apicata* and *C. jellisoni*, do not larviposit on the nostrils but rather in the host mouth and on the lower muzzle (Anderson, 1989). The first instar larvae then enter the mouth and move to the throat to feed. They further found that host cues such as carbon dioxide and pheromones helped to attract these females for larviposition (Anderson, 1989) (Appendix Fig. S1). This is most likely the same for *C. phobifer*.

The third-stage instar larva for *C. phobifer* are large in size (12–36 mm) (Bennet and Sabrosky, 1962). Their crowded feeding using their scythe-like mouthhooks can cause pitting, erosion, irritation, and enlargement of the pouch. After the mature larvae drop from the host, they will either burrow into the ground or pupate where they can. While in soil, the cuticle of these parasites hardens and blackens and then, dependent on temperature, the adult is ready for emergence 15–24 days later. Puparia also seem to be resistant if not even penchant for freezing as experiments have shown successful development after being put into the freezer for extended periods of time (Bennet and Sabrosky, 1962). Adults are robust, "furry", bee-like flies with black and yellow hairs (Bennet and Sabrosky, 1962). Adults do not have mouthparts and cannot feed. They live for 7–21 days dependent on environmental conditions. They can fly up to 36 mph. After mating and internal egg hatch, the female will use host cues to larviposit first instars thus completing the cycle (Bennet and Sabrosky, 1962).

It seems that even heavy nasal bot larvae infestations may not adversely affect the host (Bennet and Sabrosky, 1962). However, Johnson et al (Johnson et al, 1983) reported a large cerebral abscess in a mule deer with concurrent *C. phobifer* infestation. They hypothesized that larval migration may have played a role in bacteremia and the development of abscesses (Johnson et al, 1983). Mixed *Cephenemyia* species populations have been shown in mule deer in Texas and in white-tailed deer from Alberta (Kelley, 2009; Colwell et al, 2008). Notably, both of these papers called for field studies to better determine the potential role of *Cephenemyia* as vectors of CWD. Additional studies further discussed the potential of parasites to transmit prion disease and argued that the role of myiatic flies such as *Cephenemyia* could explain the endemicity of CWD (Lupi, 2003, 2005).

In this work, we show that *Cephenemyia phobifer* (Clark) nasal bots exposed to naturally CWD-infected, nonclinical white-tailed deer carry prion seeding activity. In addition, we show for the first time the presence of prion infectivity in naturally occurring insect parasites. We also demonstrate that nasal bots carry high prion infectivity titers and are able to spread their infectious cargo to other environmental components such as soil and plants. In summary, our results suggest that nasal bots could play a relevant role in CWD transmission and translocation.

## Results

### Presence of CWD prions in the nasal mucosa of prion-infected white-tailed deer

Nasal bots are exposed to the nasal cavity and retropharyngeal pouches of deer as part of their life cycle (Bennet and Sabrosky, 1962).

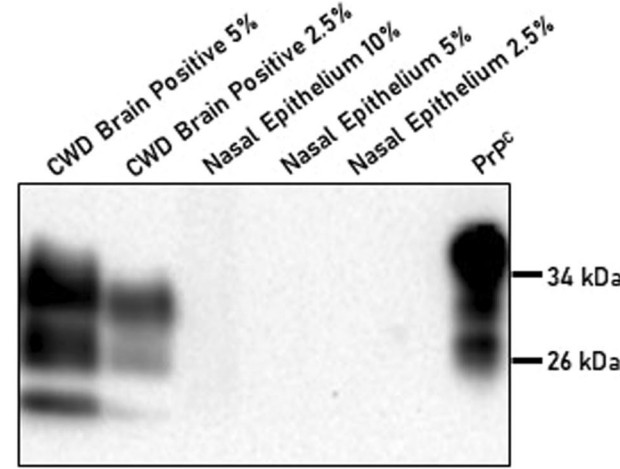

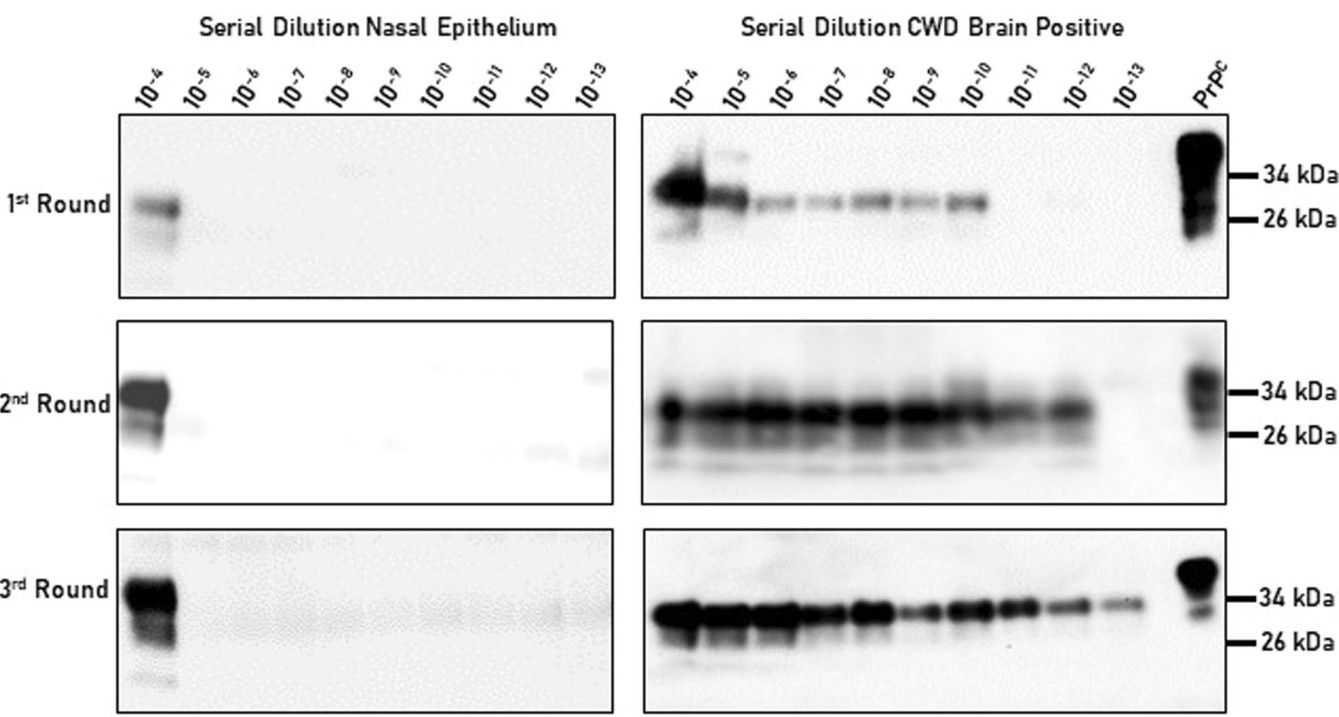

**Figure 1. Prion detection in brain and nasal epithelium from CWD-infected white-tailed deer.**

(A) The PrP$^{Sc}$ content in the brain homogenate from a clinical, experimentally infected white-tailed deer was compared to the quantities of PrP$^{Sc}$ present in the nasal epithelium of a nonclinical, naturally infected animal. Tissue homogenates were tested at different dilutions as depicted in the upper part of the panel. PrP$^{Sc}$ measurements were evaluated by western blot after PK treatment, as explained in "Methods". (B) Serial PMCA for the tissues described in (A). Tissue homogenate dilutions tested ranged from $10^{-4}$ to $10^{-13}$. Numbers at the right of each panel represented molecular weight markers. All samples were evaluated after PK digestion, except for "PrP$^{C}$" which was used to control antibody performance and electrophoretic mobility. The results presented in this panel are representative of five technical replicates. Source data are available online for this figure.

These anatomical structures are relevant for prion diseases. Specifically, humans and animals (including white-tailed deer) accumulate prions in the nasal cavity, and diagnostic methods using nasal brushes have been proposed for both species (Kraft et al, 2023; Vinny et al, 2014; Zanusso et al, 2014; Haley et al, 2016). In addition, nasal exposure to infectious prions has been proven as an efficient route of transmission (Nichols et al, 2013). In order to confirm the presence and relative quantity of CWD prions in the nasal cavity, a nasal epithelium sample (nasal mucosa mostly composed of respiratory epithelium) of a naturally infected and nonclinical white-tailed deer was interrogated for its presence of prions using western blot and PMCA (Fig. 1). The seeding activity of the nasal epithelium was

compared with that present in the brain of an experimentally CWD-infected white-tailed deer sacrificed at the terminal stage of the disease. Figure 1A shows that while prions in the brain of the CWD-infected deer were readily detected by western blot, prion signals in the nasal cavity were absent. PMCA data showed that CWD prions in the brain of the symptomatic and experimentally infected white-tailed deer were replicated with great efficiency (up to a $10^{-13}$ brain dilution), similarly as extensively described by us (Kramm et al, 2019, 2017; Bravo-Risi et al, 2021; Kramm et al, 2020). On the contrary, the seeding activity of prions in the nasal cavity seemed limited compared to the brain, as positive signals were detected only at nasal cavity homogenate concentrations up to $10^{-4}$ (Fig. 1B). Regardless, the quantities of prions present in this sample appear to be in sufficient quantities to sustain prion infectivity according to previous reports (Denkers et al, 2020). It is relevant to note that the quantity of prions in the nasal cavity of other deer may vary considerably as multiple factors such as the prion strain, route of infection, polymorphic variation of the prion protein, incubation period, among others may alter prion shedding, load, and tropism. Future experiments should define the prevalence of prions in this tissue, although extrapolations can be safely made based on a recent study evaluating prion shedding through nasal secretions (Kraft et al, 2023). Regardless, these results confirm the presence of CWD prions in the nasal cavity of a naturally infected, nonclinical white-tailed deer, and support the idea that this anatomical structure contributes to the shedding of infectious prions to the environment via secretions.

Importantly, prion deposits were also observed in the pharyngeal pouches of nonclinical, CWD-infected white-tailed deer using IHC analyses (Appendix Figs. S2 and S3). This further supports the idea that nasal bots are exposed to CWD prions during the different stages of their life cycle.

## Nasal bots contain prion seeding activity

Considering the presence of infectious prions in the nasal cavity of naturally infected, nonclinical white-tailed deer, and the abundance of nasal bots in these animals, we collected several specimens of these common deer parasites from CWD-free and -infected deer and tested them for the presence of CWD prions. A total of 34 specimens were collected from deer originating from different farmed cervid facilities across the United States (Appendix Table S1). Site MW-1 had a very low herd prevalence of CWD (3%) and the bots utilized in this study were from deer found to be CWD non-detect by immunohistochemistry (IHC) in the medial retropharyngeal lymph node. Considering this, the parasites collected at this site were used as negative controls. Whole bot specimens were collected opportunistically *postmortem* from deer euthanized as part of herd Federal/state depopulation events. Samples were homogenized at 10% w/v in PBS and tested by western blot and PMCA. As expected, proteinase K (PK)-resistant disease-associated prions (PrP$^{Sc}$) were not observed in nasal bots from any deer found to be non-detect by IHC; however, all bots collected from nonclinical CWD-positive deer displayed PMCA activity (Fig. 2; Appendix Table S1). Importantly, all specimens collected from the MW-2 and TX-1 sites displayed positive PrP$^{Sc}$ signals in either the second or third PMCA rounds. A nasal mucosa from a nonclinical, CWD-positive animal collected at the MW-2 site was used as a positive control for the PMCA assay. This sample provided positive results for prion detection in the first PMCA round, confirming our

previous findings described in Fig. 1. Importantly, no differences in the electrophoretic mobility of the PMCA products were observable, suggesting that the CWD agent present did not undergo obvious adaptation in the parasite. The lack of seeding activity in bots from the non-detect deer at the MW-1 site further validated the specificity of our assay.

Considering that bots do not express the physiological form of the prion protein (PrP$^{C}$), and that these parasites most likely acquired the infectious agent by direct contact, we wanted to determine whether the PMCA activity was restricted to the cuticle of the parasite or also present in its inner structures. To determine this, two parasites from the TX-1 and three from the MW-2 sites were dissected in order to separate their cuticles from their inner structures, as shown in Appendix Fig. S4. Our results show that CWD prion seeding activity was detected in both the outer and inner structures, suggesting that prions distribute in both parts of the developing parasite (Fig. 2). This is consistent with bots being in contact with tissues accumulating infectious prions, but also with the fact that these parasites mostly feed from the secretions released by these tissues (Bennet and Sabrosky, 1962). In fact, the internalization of disease-associated prions by bots was further confirmed by immunohistochemical analyses demonstrating the presence of CWD prions in their digestive tracts (Appendix Fig. S2).

To further explore the potential impact of these parasites in the natural spread of CWD, we collected several bot specimens from free-ranging animals in Val Verde County, Texas. This area has been recently reported to contain CWD-infected free-ranging white-tailed deer, albeit in a low prevalence (TPWD, personal communication). Between January and February 2020, 62 bot specimens were collected from nonclinical, free-ranging animals and tested by PMCA (Appendix Table S1). This preliminary study showed no PMCA activity in any of the bots analyzed which is consistent with the lack of CWD prion detection by ELISA and PMCA in the retropharyngeal lymph nodes of these animals.

## Nasal bots carry relevant quantities of CWD prion infectivity

To determine if nasal bots carry prion infectivity, we utilized a bioassay where bot homogenates were intracerebrally inoculated in tg1536$^{+/-}$ mice. These mice overexpress a cervidized version of the prion protein and are susceptible to CWD infectivity after intra-cerebral exposure to contaminated materials (Browning et al, 2004). Bot specimens for the bioassays were selected based on their PMCA status (positive or negative). Specifically, two whole bot homogenates from the MW-2 site were chosen as they provided positive signals after two PMCA rounds. In addition, the inside and outside parts of a bot collected in the TX-1 site were also selected to assess whether infectivity differs between bot components. A bot from the MW-1 site, displaying negative CWD seeding activity in PMCA, was selected as a negative control. Positive control for this bioassay included the brain extract of an experimentally infected, clinically-ill white-tailed deer of known seeding activity (Fig. 1). Challenged tg1536$^{+/-}$ mice were classified as "CWD-infected" if positive for any of the three following criteria: (i) prion clinical signs and PrP$^{Sc}$ signals in their brains after western blotting; (ii) absence of prion clinical signs and positive PrP$^{Sc}$ signals in their brains after western blotting; or (iii) absence of prion clinical signs and positive PrP$^{Sc}$ signals in their brains after a single PMCA

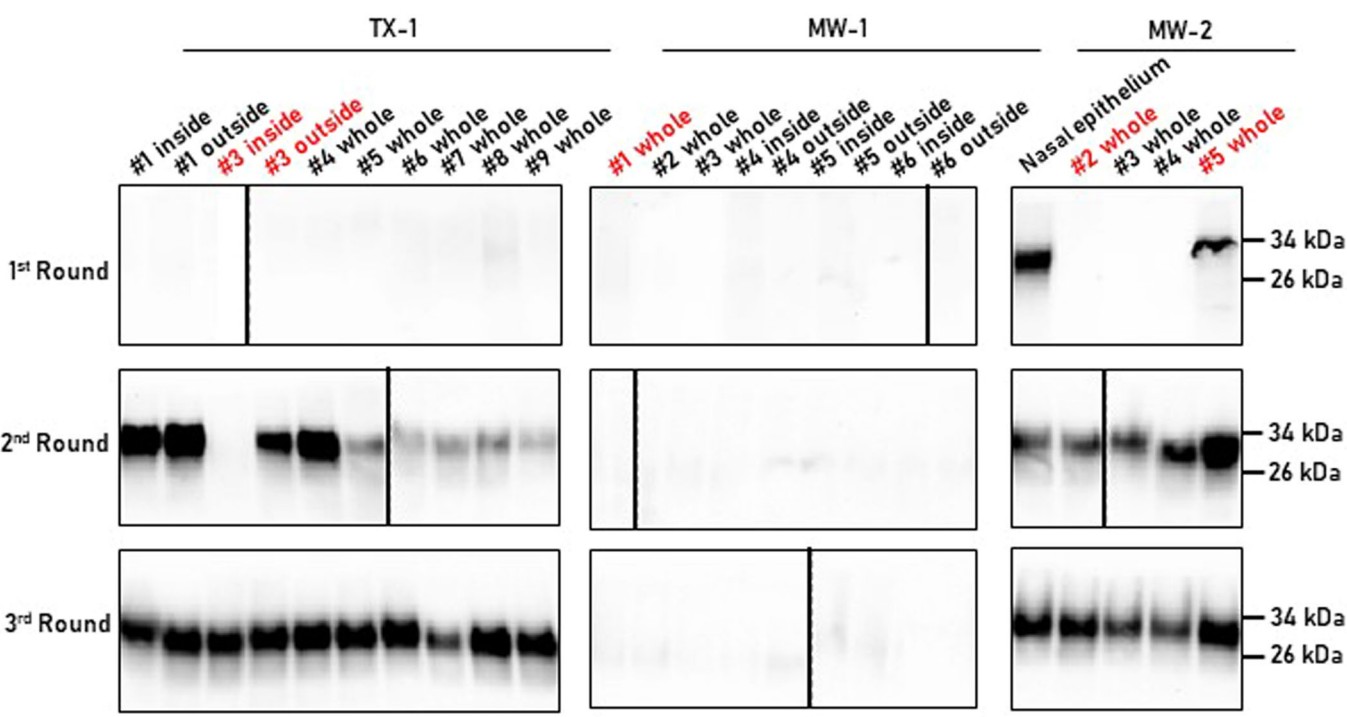

**Figure 2. PMCA analysis of nasal bots collected from farmed white-tailed deer.**

Nasal bots were obtained from farmed white-tailed deer as part of depopulation procedures performed by the United States Department of Agriculture (USDA). Bots included in this panel were collected from three different sites (TX-1, MW-1, and MW-2). Some bots were dissected to differentially assess their prion content in their protective shells (outside) and their inner structures (inside). The nasal epithelium of a CWD-positive deer from the MW-2 farm was used as a positive control. Samples marked in red were further used for bioassays. All samples depicted in this figure were tested for their PrP$^{Sc}$ content after PK digestion. The numbers at the right of each panel represent molecular weight markers. The bot specimens tested in this experiment were submitted to the PMCA analysis at least twice. Source data are available online for this figure.

**Table 1. Summary of relevant data associated with CWD-prion bioassays in tg1536 mice.**

| Inoculum | Mean DPI ± SD* | Clinical signs/inoculated (attack rate %) | PrP$^{Sc}$ signal/inoculated (attack rate %) |
|---|---|---|---|
| #5 Whole MW-2 | 477.7 ± 76.9 | 7/9 77.7% | 7/9 77.7% |
| #2 Whole MW-2 | 398, 458 | 2/8 25.0% | 5/8 62.5% |
| #3 Inside TX-1 | 550, 525 | 2/6 33.0% | 2/6 33.3% |
| #3 Outside TX-1 | 434.7 ± 18.9 | 7/7 100.0% | 7/7 100.0% |
| #1 Whole MW-1 | | 0/7 0.0% | 0/7 0.0% |
| CWD brain positive | 313.4 ± 13.3 | 7/7 100% | 7/7 100.0% |

*DPI* days post inoculation, *SD* standard deviation.
*These data consider mice sacrificed at the terminal stage of prion disease as explained in "Methods".

round. The status of each animal group in this bioassay is summarized in Table 1. Our results identified the brain material of the experimentally infected white-tailed deer as highly infectious as it induced clinical prion disease and prion propagation in 100% of the treated mice with a mean incubation period of 320 days post injection (Fig. 3A and Table 1). Our results also demonstrate the specificity of this bioassay as all mice injected with the PMCA negative bot specimen reached the experimental endpoint (600 days post injection) without showing prion-associated clinical signs and did not display PrP$^{Sc}$ signals by western blot or PMCA (Fig. 3A,B). Importantly, the whole bot homogenates collected at the MW-2 site induced prion disease in tg1536$^{+/-}$ mice albeit with incomplete attack rates (Table 1

and Fig. 3A). Interestingly, when the outer and inner part of the TX-1 site bot were tested, mice inoculated with outer bot parts had a substantially higher attack rate and much shorter incubation period compared to mice inoculated with inner bot parts (Fig. 3A and Table 1). These results were consistent with the earlier detection of prion seeding activity found in the outer structures of bots using PMCA (Fig. 2). Altogether, this information confirms that nasal bots carry CWD prion infectivity, and that much of this infectivity is present in the outer part of these parasites.

Additional analyses were made to characterize prion pathology in experimental and control mice (experimental strategy summarized in Appendix Fig. S5). Mice displaying prion-associated clinical

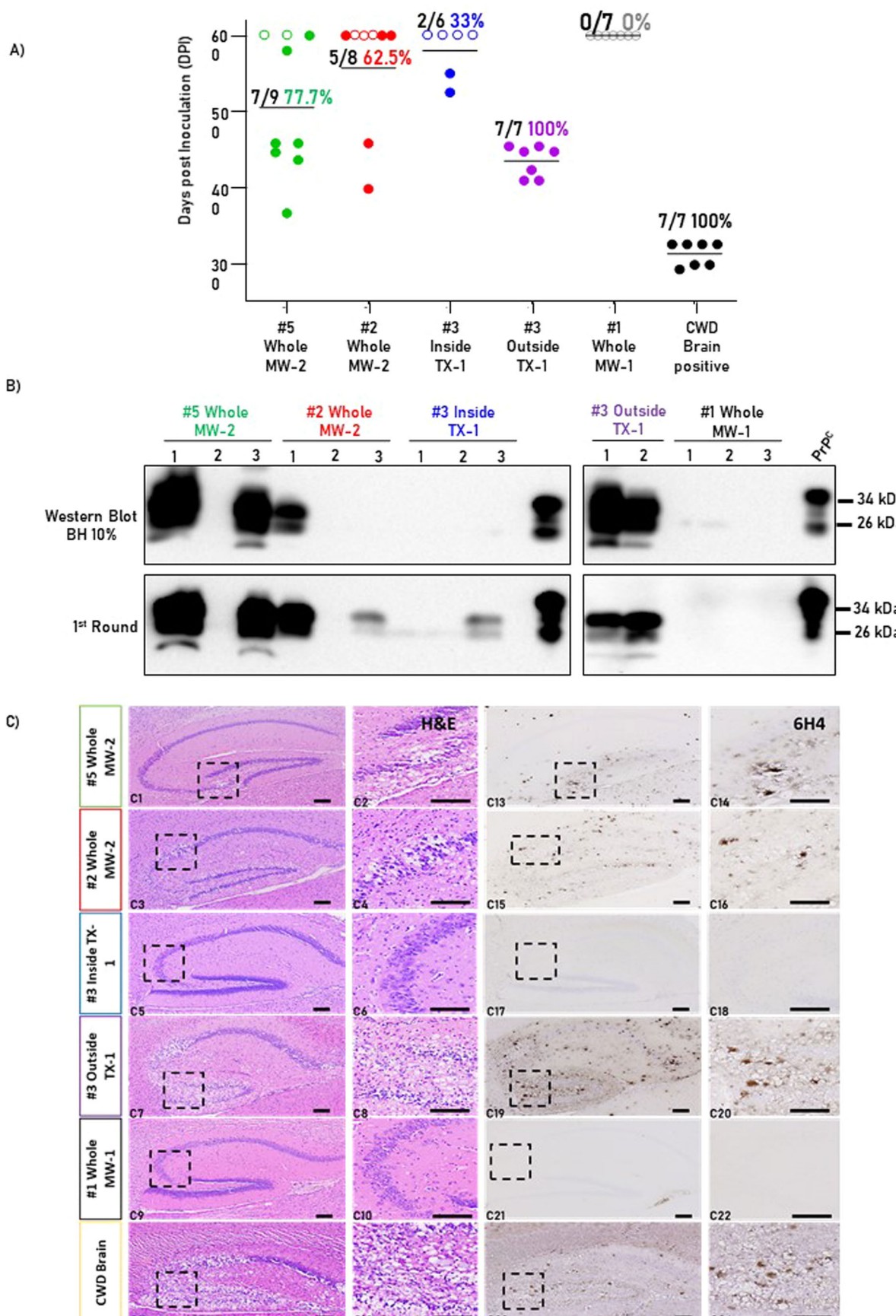

Figure 3. Infectivity bioassays to evaluate the presence of CWD infectivity in naturally occurring nasal bots.

(A) Incubation periods to terminal stages of prion disease or experimental endpoint (600 days post injection). Open circles represent animals negative for CWD prion transmission, and closed circles represent animals fulfilling the prion transmission criteria explained in "Methods". Horizontal lines in each experimental and control group represent the incubation period average. Numbers at the top of each group note the number of positive animals divided by the total number of animals in each group, and the respective attack rate (in %). (B) Representative western blot (upper panels) and PMCA (lower panels) analyses of representative brain samples from mice in experimental and control groups. Numbers in black at the top of each panel represent different animals. Numbers at the right of each panel represent molecular weight markers. All samples were evaluated after PK digestion except for "PrP^C" which was used to control antibody performance and electrophoretic mobility. (C) Representative pictures of brain sections (hippocampus) from mice in experimental and control groups. (C1–C12) represent hematoxylin–eosin to evaluate spongiform degeneration, and (C11–C24) correspond to immunohistochemistry (IHC) detecting prion protein deposits. Specifically, this figure shows representative brains from mice displaying clinical signs at the groups "#5 Whole MW-2", "#2 Whole MW-2", "#3 Outside TX-1" and "CWD brain". This figure also displays a mouse from the "#3 Inside TX-1" group that was not clinically infected by CWD prions. This brain did not display spongiosis or PrP accumulation, similar to the brains of mice in the negative control group ("#1 Whole MW-1"). Micrographs were acquired at two different magnifications as depicted in "Methods". Dotted squares represent areas of interest that were magnified and shown at the right. Bars represent 100 μm. Between six and nine animals per group were used in this bioassay. Source data are available online for this figure.

signs showed pathological features of prion disease in their brains, including extensive spongiform degeneration and accumulation of PK-resistant prion protein (Fig. 3C). In addition, animals determined to be negative for prion replication did not show these pathological hallmarks (Fig. 3C; Appendix Fig. S6). Gross-examination of these specimens in different brain regions did not show differences in the anatomical distribution of either spongiform degeneration or prion deposition, suggesting that the CWD prions present in all nasal bots corresponded to the same prion strain (Fig. 3C; Appendix Fig. S7). This was confirmed by western blotting and PMCA where no differences in the PK-resistant form of PrP^Sc were found (Fig. 3B). It is important to note that all mice not displaying prion-associated clinical signs were sacrificed at the 600 days post inoculation experimental endpoint. In the "#5 Whole MW-2" group, one mouse was sacrificed at the endpoint but did display non-terminal prion-associated clinical signs. As expected, this animal had detectable PrP^Sc levels in its brain after western blot analysis. In the "#2 Whole MW-2" group, just two mice displayed clinical signs and were sacrificed accordingly before the experimental endpoint (Fig. 3A,B). All remaining mice from this group were sacrificed without clinical signs 600 days after inoculation. Importantly, three of them showed detectable levels of CWD prions, indicating sub-clinical prion infection.

## Nasal bots accumulate infectious prions

As mentioned, the mouse bioassays demonstrated that nasal bots carry significant quantities of prion infectivity. This information suggests that nasal bots can take transmission-relevant levels of infectious prions regardless of their exposure to environments containing limited quantities of infectivity. To further explore this, nasal bots collected from white-tailed deer non-detect for CWD (determined by *postmortem* immunohistochemical analyses of the retropharyngeal lymph node) were exposed to serial dilutions of a CWD-positive brain extract of known seeding activity. Brains from CWD-infected tg1536 mice were used for these studies considering the large quantities of CWD prions needed for these analyses and their availability in our laboratory. Considering this, and to evaluate the capacity of nasal bots to bind CWD prions, parasites collected from CWD non-detect deer were transiently exposed to a tg1536 CWD-laden brain extract at different times ranging from 1 s to 1 h (Fig. 4A). After 1 PMCA round, we retrieved PrP^Sc signals in bots from all the time points, although the group of bots exposed for the longest time showed a higher frequency of positivity in the different replicates tested (Fig. 4B). In a second PMCA round, all

bots from all groups displayed positive signals. These results were maintained in a third PMCA round (Fig. 4B). As expected, bots exposed to a CWD-free tg1536 brain extract did not show seeding activity in PMCA (Fig. 4C). These data suggest that bots can bind prions even after transient exposure to CWD prions.

In addition, we performed a second experiment in which bots collected from CWD non-detect deer were exposed overnight to different concentration of prions. To better appreciate the ability of bots to interact with CWD prions, the exposed parasites were homogenized and serially diluted before being tested in the PMCA assay (Fig. 5A). In this experiment, we observed that bots exposed to the $10^{-3}$ and $10^{-5}$ CWD-laden brain extracts displayed positive signals even after bot extracts were diluted $10^6$ times (Fig. 5B). Interestingly, bots exposed to a $10^{-8}$ CWD brain extract were detected only in its lower dilution ($10^{-2}$). It is relevant to note that a $10^{-8}$ CWD brain extract approaches to the range of the limiting detection by PMCA (Fig. 5C), suggesting that nasal bots are able to bind small quantities of prions. As expected, bots incubated with a prion-free brain extract did not provide positive signals (Fig. 5D). A summary of prion detection obtained in this experiment is presented in Fig. 5E. Overall, these data suggest that nasal bots are able to accumulate prion infectivity from their environment, regardless of the number of prions to which they are exposed.

## Interaction of CWD-contaminated bots with soils and plants

Nasal bot larvae exit deer through the nasal cavity and drop to the ground as part of their life cycles. While in soil, these parasites complete their development into flies (Bennet and Sabrosky, 1962). Importantly, a considerable proportion of nasal bots do not complete their cycles and remain in soils until they die naturally or by environmental stimuli (Colwell et al, 2008). In addition, nasal bots developing into flies will shed their heavily CWD-contaminated protective shells that will remain in the soil. Considering (i) our previous results demonstrating that nasal bots accumulate infectious prions, and (ii) the important role of the environment in CWD transmission (reviewed in (Escobar et al, 2020; Bartelt-Hunt and Bartz, 2013)), we modeled a potential scenario in which nasal bots enter the soil and contaminate both this specific matrix and the plants that grow on it (Fig. 6A). To simulate this scenario, bot homogenates previously tested as CWD-positive by PMCA were pooled and mixed with a commercially available compost soil. As a negative control, the same procedure was repeated but using bot homogenates displaying negative

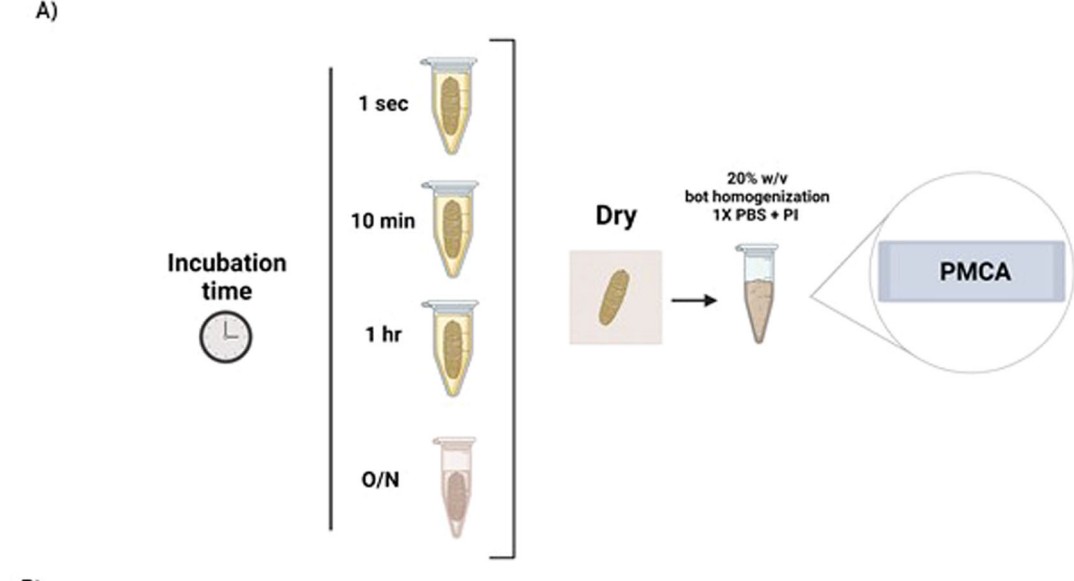

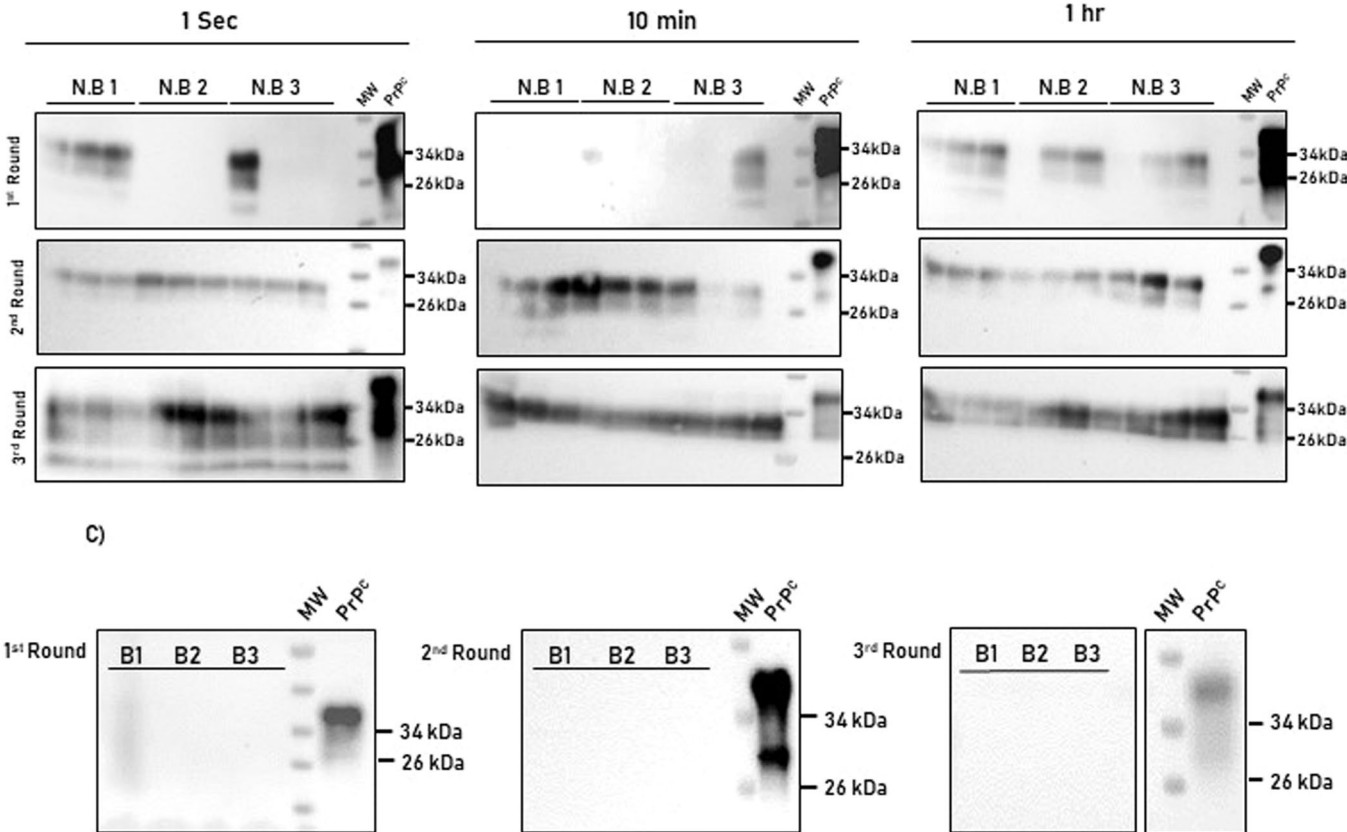

in vitro prion seeding activity. As shown in Fig. 6B, 2/5 replicates of the soil exposed to CWD-positive bot homogenates provided positive PMCA signals. Not surprisingly, the soils mixed with the CWD-negative bots did not display in vitro prion seeding activities. This result suggests that nasal bots have the potential to contaminate soils in natural settings.

Additional fomites associated with environmental prion transmission include plants. Although CWD prion detection has not been reported in naturally exposed plant specimens, it has been demonstrated that they can uptake infectious particles from soil under experimental conditions (Pritzkow et al, 2015). To explore whether soil contamination with CWD through nasal bots leads to prion uptake by

Figure 4.   Time-dependent adsorption of CWD prions by nasal bots.

(A) Experimental strategy depicting the CWD- (top three) and non-infected- (bottom) brain extracts used to expose nasal bots at different times (1 s, 10 min, and 1 h). "O/N" depict the negative control that consisted in the exposure of bots to CWD-free tg1536 brain extracts overnight. Samples were prepared and analyzed using PMCA as described in "Methods". This panel was created using Biorender. (B) PMCA analysis of the nasal bots exposed to brain homogenates for different times. Results of the first (top panels), second (middle panels), and third (bottom panels) are displayed. As shown in the blots, three bots (N.B. 1–3) were used in each group. Each bot was tested three times to assess the reproducibility of the results. (C) Blots depicting PMCA analyses of bots incubated overnight with the CWD-free brain extract. Numbers at the right of each panel represent molecular weight markers. All samples were evaluated after PK digestion with except for "PrP$^C$" wich was used to control antibody performance and electrophoretic mobility. MW: lane containing a molecular weight marker ladder. Experiments were performed using three biological replicates and three technical replicates. Source data are available online for this figure.

plants, the same soils studied above were used to grow wheatgrass (Fig. 6A). Plant specimens were harvested after 31 days, and leaf and root homogenates were evaluated for potential prion uptake. Our results show that while no signs of prion seeding activity were identified in leaves from plants exposed to soils contaminated with CWD-positive bots, clear signals were observed in the roots (Fig. 6C). The specificity of this assay was validated by equivalent analyses in plant specimens grown in soils exposed to CWD-negative nasal bot homogenates that did not display PMCA seeding activity. These results further demonstrate the potential role of CWD-exposed nasal bots in environmental contamination through grass plants which are common sources of deer feed. In summary, the information presented in this study proposes these parasites as contributors in the environmental spread of CWD prions.

## Discussion

Several mechanisms have been proposed to explain the natural spread of CWD. These include direct animal contact, vertical transmission, and exposure to contaminated environmental fomites (Escobar et al, 2020; Bartelt-Hunt and Bartz, 2013; Zabel and Ortega, 2017). The latter is perhaps the best studied and most critical for the long-term maintenance of CWD within a population. A wealth of data suggests that CWD prions deposited in the environment through excreta, animal tissues (e.g., placenta (Bravo-Risi et al, 2021), or carcasses (Miller et al, 2004; Escobar et al, 2020)) can persist and remain infectious for extended periods of time (Mathiason et al, 2009). Urine, feces and saliva from infected deer have been identified as the main sources of environmental prion contamination as they spread small but consistent quantities of CWD prions throughout the course of the disease (McNulty et al, 2020; Henderson et al, 2015; Plummer et al, 2017). Infectious prions released from infected animals through excreta progressively attach to different environmental fomites, including soils (Johnson et al, 2006; Plummer et al, 2018), plants (Pritzkow et al, 2015) and other natural and human made environmental components (Pritzkow et al, 2018) that may serve as reservoirs of disease-associated prions. Whether parasitic insects interacting with diseased animals participate in these events has not been extensively explored.

In this study, we evaluated nasal bots fly larvae as potential vectors of CWD transmission. We decided to explore the prion content in these parasites for several reasons including (i) their common occurrence in cervid populations; (ii) their interactions with tissues described to contain prion infectivity; (iii) the recent description that prions are shed by nasal secretions before accumulation in lymphoid tissues (Kraft et al, 2023); and (iv) their further interaction with soils after exiting their hosts. The results

presented here show that PMCA seeding activity is readily identified in nasal bots collected from naturally infected, non-clinical white-tailed deer. Importantly, these parasites contained transmission-relevant levels of CWD prions as confirmed by mouse bioassays. Our results also indicate that prion infectivity in these parasites was mostly present in their protective shells. To explore the overall infectious potential of nasal bots, we estimated the infectivity titers of bot homogenates based in previously published data (Denkers et al, 2020). In light of our results, we estimate that the infectious potential of 10 μL of the 10% w/v bot shell homogenate used in this study was equivalent to a $10^{-3}$ brain dilution from experimentally infected tg1536$^{+/-}$ mice sacrificed at the terminal stage of the disease (402.6 ± 38.2 days post injection; $n = 8$). It is important to mention that tg1536$^{+/-}$ mice overexpress the white-tailed deer PrP$^C$, so PrP$^{Sc}$ quantities in the brains of these mice are considerably higher than those found in naturally infected cervids (approximately ten times considering the materials used in this study, Appendix Fig. S8). Regardless, it is safe to state that nasal bot homogenates are between 10 and 100 times less infectious than the brains of clinical, experimentally infected deer. An average nasal bot weight 0.8 mg, and previous reports demonstrate that just 100–300 ng of CWD-infected brain material are enough to experimentally infect deer by the oral route (Denkers et al, 2020). Considering that the most plausible way of exposure of naive deer to CWD-contaminated nasal bots is through ingestion (discussed below and Fig. 7), we estimate that a single parasite should be enough to infect at least one deer. In that sense, nasal bots exposed to CWD-infected deer could be visualized as "capsules" of infectious prions that may play a previously unexplored but relevant role in disseminating this disease. An additional aspect relevant for CWD environmental contamination is that a single deer can harbor more than one hundred of these parasites at a single time (TPWD field personnel, personal communication). Consequently, the opportunity for nasal bots to substantially augment overall CWD prion load within the environment is high.

As mentioned above, nasal bots exit their hosts through the nasal cavity after completing their developmental stages in the pharyngeal pouches. These parasites then move on in their life cycles in soils from where they finally emerge as mature flies (Bennet and Sabrosky, 1962). It is estimated that a substantial percentage of bots deposited in soils do not complete their cycles into flies as they die for multiple reasons. Bots emerging from CWD-afflicted deer will deposit their infectious prion load in the soil, contributing to environmental contamination. For the remaining bots that fully complete their developmental cycle, their highly infectious shells will remain in soil and still contaminate the environment. We believe that prions released by bots will bind to soils where they will persist for extended periods. Considering the

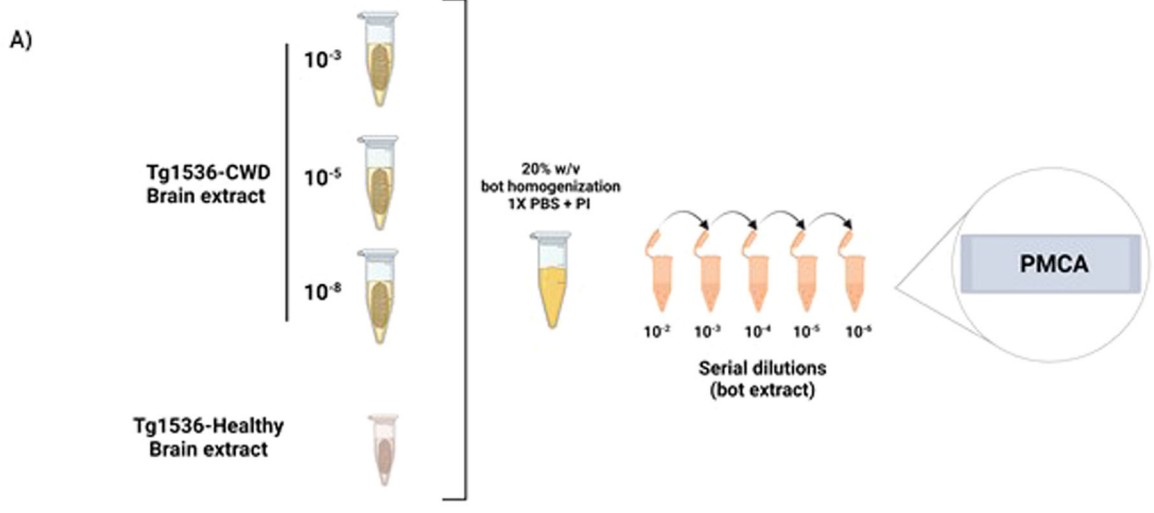

A)

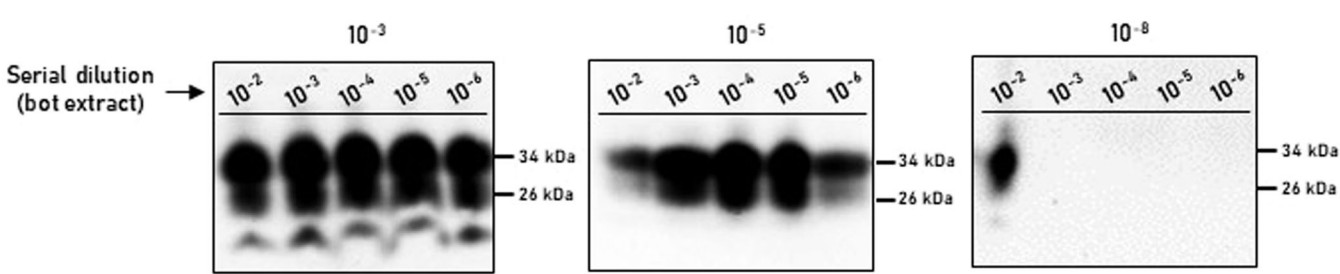

B) Tg 1536-CWD Brain extract

Serial dilution (bot extract)

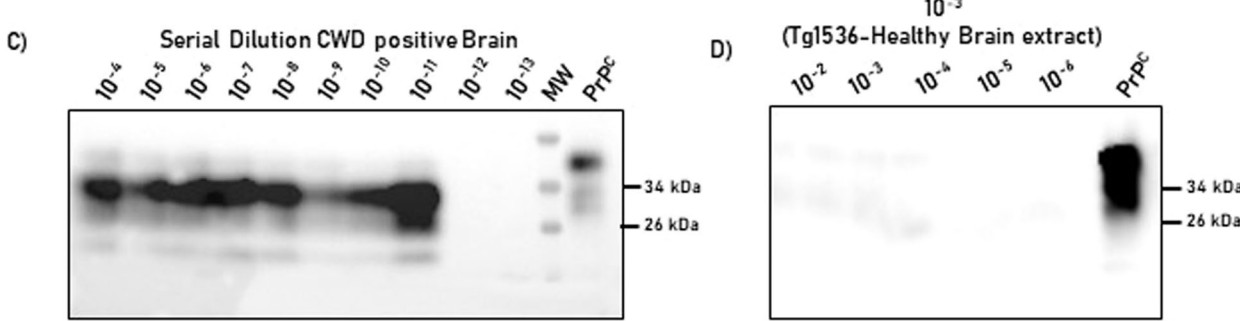

C) Serial Dilution CWD positive Brain

D) 10⁻³ (Tg1536-Healthy Brain extract)

Serial dilution (bot extract)

E)

| Tg1536-CWD Brain extract | $10^{-2}$ | $10^{-3}$ | $10^{-4}$ | $10^{-5}$ | $10^{-6}$ |
|---|---|---|---|---|---|
| $10^{-3}$ | | | | | |
| $10^{-5}$ | | | | | |
| $10^{-8}$ | | | | | |

**Figure 5. Concentration-dependent adsorption of CWD prions by nasal bots.**

(A) Experimental strategy depicting the CWD- (top three) and non-infected- (bottom) brain extracts used to expose nasal bots at different concentrations. After brain extract exposure, bots were homogenized and serially diluted before being tested in PMCA. This panel was created using Biorender. (B) Representative blots showing PrP$^{Sc}$ content in PMCA products (3rd round) of serially the diluted bot homogenates. (C) PMCA analysis of the brain extract used to contaminate bots. (D) Representative blot showing the absence of PrP$^{Sc}$ signal in PMCA products (3rd round) of bots exposed to the CWD-free brain extract. Numbers at the right of each panel represent molecular weight markers. All PMCA products were evaluated after PK digestion. (E) Summary of detection prion detection using PMCA for the results illustrated in (B). Experiments were performed using three biological and three technical replicates. Source data are available online for this figure.

prevalence of these parasites, it is plausible that deer will ingest CWD-contaminated whole, decaying or pulverized bots or bot's shells while grazing together with soil; thus, bots could act as CWD vectors. In addition, we demonstrated that grass plants grown in soils exposed to CWD-contaminated bots can incorporate prions into their roots, expanding the chances for these parasites to infect naive deer, considering that they may ingest these plant components when other food sources are scarce. We believe that CWD transmission mediated by nasal bots are more likely to occur in captive settings where the density of animals is high, and the cumulative prion load within the environment can increase more rapidly. The potential of these parasites to contribute to environmental contamination in comparison with excreta such as urine and feces is especially high when considering their ability to accumulate relevant quantities of CWD prions. The different possibilities in which nasal bots may contribute to CWD prion transmission in natural conditions are summarized in Fig. 7.

Unfortunately, this study was unable to link the stage of the incubation period in deer with the CWD infectivity present in bots. We were also unable to compare the infectivity of bot-donor deer tissues with those found in the parasites. These paradigms are relevant as they may provide useful information related to the temporal shedding of prions through these parasites. Future studies in this direction should provide us with valuable data to address the true potential of nasal bots in CWD dissemination. Nevertheless, this study further supports the major role of these parasites in transmitting CWD either directly, or through environmental components such as soils and plants.

Considering the biology of nasal bots, and the ability of these parasites to accumulate prion infectivity, there are several questions that need to be addressed in future studies. The most obvious and relevant is whether a mature fly emerging from CWD-contaminated nasal bots carries CWD prions. If so, is the CWD agent present in the deposited larvae? If positive, are prions present in larvae at quantities enough to sustain disease transmission? If CWD prions persist throughout the life stages of this common parasite, new strategies for CWD disease management will undoubtedly have to be created or at least considered. In addition, we know that nasal bots comprise a variety of subspecies, each interacting with specific host populations in different ways (e.g., geographic locations, seasons). Studying the overall prevalence of nasal bot infections and the interaction of botfly subspecies with their respective hosts should be of primary focus in future research seeking to elucidate CWD transmission. Furthermore, it is known that the interaction between prions and environmental components will vary depending on several factors including the specific prion strain(s), and the composition of the environmental fomites (soil, plants, different surfaces, etc.). These interactions should also be the focus of future research linking parasites and the environmental dissemination of naturally occurring prions.

The findings described in this article contribute to the complexity of CWD transmission on the landscape, and demonstrate that there is much remaining to be explored regarding the relevance of nasal bots and other parasites in the CWD epidemic. Further characterizing these relationships will likely add new challenges to CWD management, but this knowledge will prove invaluable to those tasked with conserving CWD-susceptible cervid populations for generations to come.

## Methods

### CWD brain sample

The white-tailed deer brain sample used in this study was collected from an experimentally infected animal. This sample was kindly donated by Dr. Edward Hoover (Colorado State University, USA). The donor animal displayed prion-associated clinical signs at the moment of euthanasia and was further confirmed as prion-infected using biochemical methods. This brain tissue was homogenized at a concentration of 10% w/v (weight/volume) in phosphate-buffered saline (PBS) (Hyclone PBS, GE Healthcare Life Sciences) supplemented with a protease inhibitor cocktail (without EDTA, Roche). Homogenization was performed in a Precellys® 24 homogenizer using Soft Tissue Homogenizer tubes (Bertin Corp. CK14).

### Nasal epithelium samples

The nasal epithelium samples were opportunistically obtained from nonclinical, naturally CWD-infected, captive white-tailed deer as part of CWD depopulation procedures performed by the USDA. These samples were homogenized at 10% w/v in PBS supplemented with a protease inhibitor cocktail as mentioned above. Homogenates were prepared using 2 mL Hard Tissue Homogenizer tubes (Bertin Corp. CK28) in a Precellys® 24 homogenizer.

### Larvae samples

Nasal botfly larvae (*Cephenemyia phobifer*) were collected from the nostrils and pharynx of white-tailed deer. Species identification was performed by USDA Parasite Identification Lab at the National Veterinary Services Laboratory (NVSL). Bots were opportunistically collected from two sources: (i) captive white-tailed deer as part of CWD depopulation procedures performed by the USDA; and (ii) free-range white-tailed deer trapped by the Texas Parks and Wildlife Department (TPWD) as part of screening practices. Animals from which larvae were removed were CWD tested by the USDA NVSL or by Texas Veterinary Medical Diagnostics Laboratory via IHC or ELISA on medial retropharyngeal lymph node (MRPLN) and/or brain stem samples. Larvae collected ranged from white to yellowish-brown

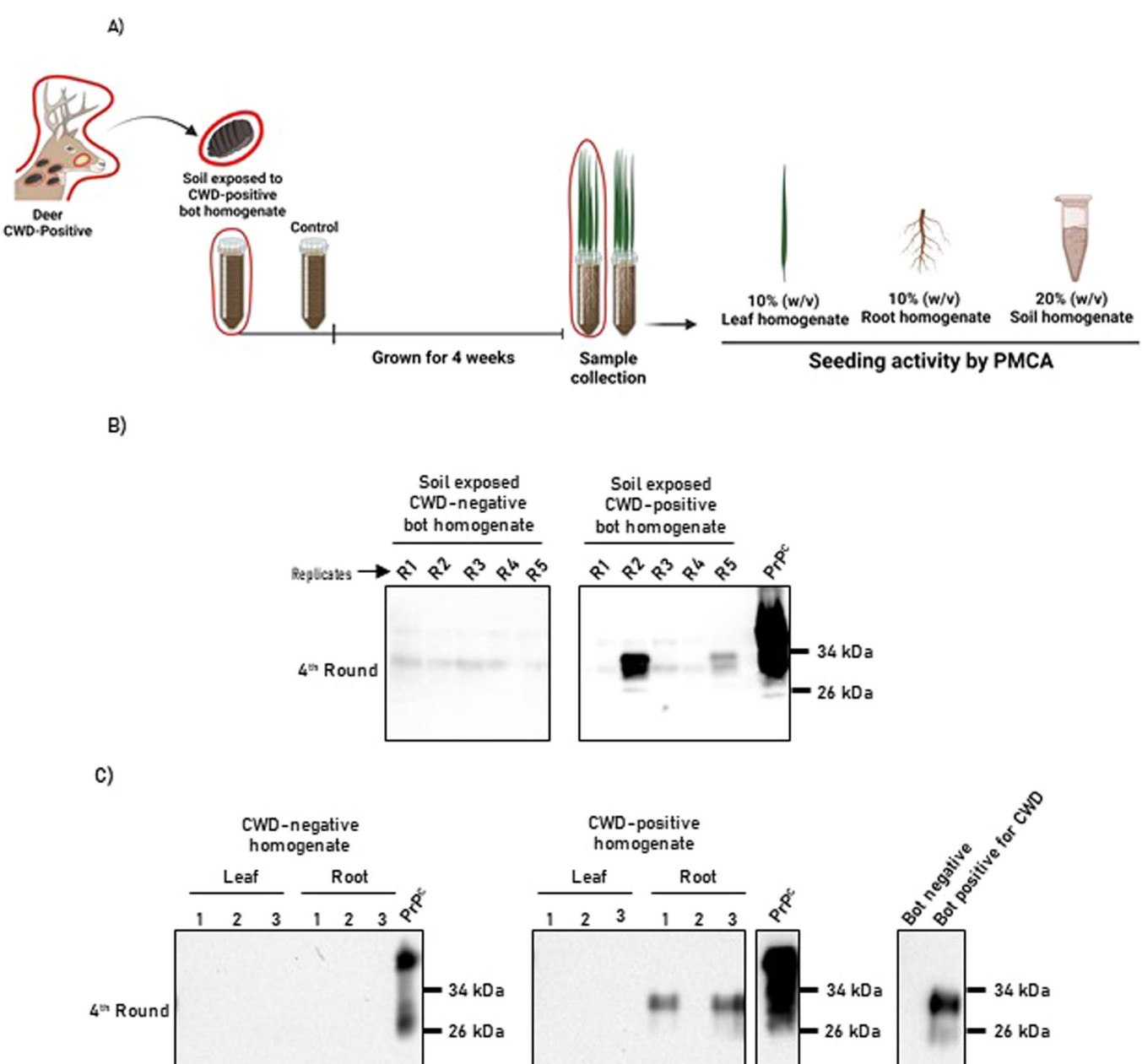

**Figure 6. Uptake of CWD prions by plants grown in soils exposed to CWD-contaminated nasal bots.**

(A) Experimental strategy depicting soil exposure to bot homogenates, plant growth and sample collection. Images in red circles represent CWD-contaminated materials. Panel created using Biorender. (B) PMCA analysis of soil samples used in this study. Numbers at the top of each panel represent different replicates for soil specimens exposed to CWD-contaminated and CWD-free nasal bot homogenates. (C) PMCA analysis of leaves and root specimens grown in soils exposed to CWD-contaminated and CWD-free nasal bots. The panel at the right represents the PMCA results of the bot extracts mixed with soils. Results depicted in (B, C) correspond to a fourth PMCA round. Numbers at the right of each panel represent molecular weight markers. All samples were evaluated after PK digestion, except for "PrP^C" which was used to control antibody performance and electrophoretic mobility. Experiments were performed using five biological replicates. Source data are available online for this figure.

in color and averaged 1–13 cm in length. Some of these parasites were carefully dissected to separate their protective shells from their inner structures. Parasites were homogenized as a whole or in their dissected parts at 10% w/v in PBS and supplemented with a protease inhibitor cocktail (without EDTA, Roche®). Homogenization was performed in a Precellys® 24 homogenizer using Soft Tissue Homogenizer tubes (Bertin Corp. CK14) for the internal part. Hard Tissue Homogenizer

tubes (Bertin Corp. CK28) were used for to homogenize whole bots and protective shells.

## Histological analyses of nasal bot larvae samples

Bots were collected and placed in 10% neutral buffered formalin for 3 days. The bots were trimmed for histology, sectioned at 5 µm,

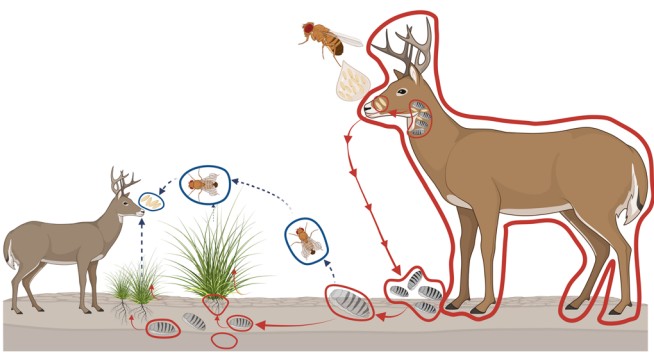

**Figure 7. Potential mechanisms of CWD-prion spread mediated by nasal bots and bot flies.**

After placing larvae in a CWD-infected deer, nasal bots develop in CWD-infected environments (nasal cavity and pharyngeal pouches). While advancing in their life cycle, nasal bots abandon their hosts and deposit in soils where they spread prion infectivity by being directly ingested by naive deer or by depositing CWD-contaminated material in the soil. Plants growing in contaminated soils may also acquire prion infectivity that will be available to deer. Deer and nasal bots enclosed in a red line represent CWD-bearing organisms. Red arrows represent the processes analyzed in the present study. The hypothetical scenario in which mature flies spread infectious prions is depicted in discontinuous arrows and flies enclosed by blue circles. Figure created using Biorender.

stained with H&E, and immunostained (IHC) with the anti-prion protein monoclonal antibody F99/97.6.1 (mAb 99) using the Ventana platform as previously described (Spraker et al, 2002). Positive and negative control slides containing lymphoid tissue and brain were stained with each run.

## Preparation of PMCA substrate

Brain extracts from homozygous tg1536 mice expressing the deer *PRNP* gene in a mouse *prnp* knock-out background (Browning et al, 2004) were used as PMCA substrate. Breeder mice were kindly provided by Dr. Glenn Telling (Colorado State University, USA) and the colony was expanded at the Center for Laboratory Animal Medicine and Care (CLAMC) at UTHealth. Mice were bred and maintained in standard conditions (e.g., water and food provided ad libitum, 12-h light/dark cycles, 23 °C, 70 ± 10% humidity) in a prion-free room. Mice were sacrificed using $CO_2$-induced asphyxiation followed by cardiac perfusion with 40 mL of cold perfusion buffer (PBS supplemented with 5 mM EDTA). Brains were collected, snap-frozen in liquid nitrogen, and stored at −80 °C until use. Brains were homogenized at a concentration of 10% (w/v) in PMCA conversion buffer (PBS supplemented with 1% v/v Triton X-100, 150 mM NaCl and protease inhibitor cocktail (cOmplete, Roche®)). Then, homogenates were centrifuged at $805 \times g$ and 4 °C for 45 s. After centrifugation, the homogenate supernatants were removed and the pellets were discarded. Supernatants were further mixed, aliquoted, snap-frozen in liquid nitrogen, and stored at −80 °C until use.

## Protein misfolding cyclic amplification (PMCA)

The PMCA procedure was performed as extensively described in our previous publications (Morales et al, 2012; Kramm et al, 2019, 2017, 2020; Bravo-Risi et al, 2021) and specifically adapted

for CWD prion amplification. PMCA reactions consisted of mixing 10 μL of a given sample with 90 μL of PMCA substrate (brain homogenates from homozygous tg1536 mice (tg1536⁺/⁺) prepared in conversion buffer as described in (Kramm et al, 2020, 2017; Bravo-Risi et al, 2021)). These mixtures were submitted to 144 cycles of incubation and sonication for the first PMCA round (each PMCA cycle consisting of 29 min and 40 s of incubation and 20 s of sonication). The resulting samples were subjected to additional rounds of PMCA (96 cycles each) by mixing 10 μL of the PMCA products of each round with fresh PMCA substrate (90 μL). PMCA products were treated with PK and examined by western blotting. As controls, each PMCA reaction set (testing approximately ten samples) included serial dilutions of a CWD brain of known PMCA activity and at least four unseeded reactions (negative controls, or NC).

## Proteinase K (PK) treatment

To assess the potential presence of disease-associated prion proteins (PrPSc), 20 μL of tissue extracts or PMCA products were treated with 100 μg/mL proteinase K (PK, Sigma-Aldrich) at 37 °C for 90 min. These reactions were conducted under shaking at 450 rpm using a tabletop Eppendorf® thermomixer. PK activity was stopped by adding LDS sample buffer and heating for 10 min at 90 °C.

## Electrophoresis and western blot

Protein electrophoresis was performed using NuPAGE 4–12% or 12%, Bis-Tris gels (Invitrogen, Carlsbad, CA, USA) with MOPS Buffer at 80 V for 20 min, and then 140 V for 1 h and 40 min. Proteins were transferred to nitrocellulose membranes (GE Healthcare Amersham) at 100 V for 60 min, using the recommended transfer buffer. Membranes were blocked using 10% (w/v) non-fat milk and probed with primary monoclonal 6H4 antibody (Prionics) diluted in a 1:12,500 ratio. A secondary antibody (polyclonal anti-mouse IgG (whole molecule)–peroxidase antibody produced in sheep (Sigma-Aldrich) diluted at 1: 3000 was used to detect PrP signals. Both antibodies were prepared in PBST—0.05% and incubated for 1 h at room temperature. The membrane was finally washed three times for ten min with PBST—0.05% and exposed to ECL (GE Healthcare Amersham) to visualize luminescent signals in a dark room (BioRad).

## Bioassay

Heterozygous tg1536⁺/⁻ mice (Browning et al, 2004) were used for the bioassay. Mice were intracerebrally inoculated at approximately eight weeks of age with 10 μL of 10% (w/v) whole bot or bot components homogenates. Injection of a white-tailed deer CWD-infected brain (preparation explained above) was used as a positive control. For injection, mice were anesthetized using isoflurane and fixed to a mouse stereotaxic frame. Unilateral injections were performed at a single point in the right hippocampal area using the following coordinates as measured from bregma: anteroposterior, −2.0 mm; mediolateral, −2.0 mm; dorsoventral, −2.0 mm. The injection was conducted at a rate of 0.5 μL/min, and the needle was left in place for 3 min before retraction. Animals were placed on a thermal pad until recovery. Then, mice were monitored for the

appearance of prion-associated clinical signs between five and seven days per week throughout the duration of the study. Clinical signs associated with prion disease in these mice include ataxia, stiff tail, head bobbing, and rough coat. The progression of clinical signs in experimental and control mice was done using a previously published scale (Castilla et al, 2008) graduated from 1 (healthy animal) to 5 (animal at terminal stage of the disease and not able to stand). Mice were sacrificed when classified at stage 4 for longer than one week, or when reaching the 600 days post injection experimental endpoint. All mice were sacrificed by $CO_2$ inhalation and death was confirmed by decapitation. The brains from these mice were collected and dissected in their two hemispheres. Half of the brain (non-injected hemisphere) was stored in Carnoy solution for 24 h and then transferred to 95% ethanol until used for histology. The other half (injected hemisphere) was stored at −20 °C until used for biochemical analyses. Animals were considered as CWD-infected if they were positive for any of the following criteria: (i) displaying prion clinical signs and PrP$^{Sc}$ signals in their brains after western blotting; (ii) absence of prion clinical signs and positive PrP$^{Sc}$ signals in their brains after western blotting; or (iii) absence of prion clinical signs and positive PrP$^{Sc}$ signals in their brains after a single PMCA round. All animal procedures were performed following NIH guidelines and approved by the Animal Welfare Committee of the University of Texas Medical School at Houston. Mice were kept at a 12-h light/dark cycle and maintained at 23 °C in an environment with 70 ± 10% humidity (Innorack® IVC Mouse 3.5). The mice were allowed to access food and water *ad libitum*.

## Biochemical analysis of PrP$^{Sc}$ in tg1536$^{+/−}$ mice

Brain samples collected from the bioassay were homogenized at 10% w/v in PBS (GE Healthcare Life Sciences) supplemented with a protease inhibitor cocktail (without EDTA, Roche) using a Precellys® homogenizer. Tissues were homogenized in Soft Tissue Homogenizer tubes (Bertin Corp. CK14). Analysis of PrP$^{Sc}$ in these tissues was evaluated by western blotting after PK treatments in the same manner as explained above.

## Neuropathology

After fixation in Carnoy solution and 95% ethanol, the brains were processed by dehydration and embedding in paraffin to be sliced on a Leica® microtome (HistoCore AUTOCUT). Sagittal brain sections of 10 μm were obtained and placed on charged glass slides and air-dried overnight at room temperature. The next day, the slides were placed in an oven at 60 °C for 20 min and then deparaffinized and hydrated. For the histological evaluation and presence of spongiform degeneration, hematoxylin and eosin (H&E) staining was performed using a standard protocol. Analysis of PrP$^{Sc}$ deposits was performed by IHC staining. For IHC, the deparaffinized and hydrated sections were incubated with PK solution (10 μg/mL) for 10 min at room temperature. Then, the brain sections were incubated with 4M guanidine thiocyanate for 1 h at room temperature. Endogenous peroxidase activity was blocked with a 3% $H_2O_2$/10% methanol solution prepared in PBS. Non-specific antibody binding was minimized by incubating the slides with 3% BSA/0.2% Triton X-100 prepared in PBS. The staining of the PrP$^{Sc}$ aggregates was performed using the 6H4

antibody (1:250) incubated overnight and then probed with sheep anti-mouse IgG-Horseradish Peroxidase (HRP) secondary antibody (1:500) for 1 h. The reaction was visualized using the chromogen 3,3'-diaminobenzidine (DAB) Kit with Nickel (Vector Laboratories), following the manufacturer's instructions. Sections were counterstained with 20% hematoxylin, dehydrated in graded ethanol, cleared in xylene, and covered with DPX mounting media (Innogenex). Pictures were taken with a Leica DMi8 microscope (Leica Microsystems) at ×10 and ×40 magnifications.

## Evaluation of the brain PrP$^{Sc}$ content in a CWD-infected white-tailed deer and CWD-infected tg1536 mice

Serial dilutions of deer and mice brains (1.25% and 0.6% w/v) were performed in PBS. Then, extracts were subjected to PK treatments as described above to visualize disease-associated prion proteins. PK-digested samples were then examined by western blotting as described above. The BioRad ImagenLab program was used to measure densitometric signals that were normalized to the highest signal within each membrane (1.25% tg1536 brain extracts). The data were analyzed using the GraphPad Prism program. The experiments were performed in three technical replicates.

## Nasal bot—CWD prions interaction at different times

A CWD-laden brain extract ($10^{-3}$ brain dilution) was prepared from clinically sick CWD-infected tg1536 mice in PBS supplemented with a protease inhibitor cocktail as described above. This brain extract was used to expose nasal bots to infectious prions. The nasal bots used in this study were collected from different CWD non-detect white-tailed deer. The bots were washed with PBS before mixing them with the brain extract to remove traces of blood and mucosa. The prion adsorption procedure was performed by placing bots in a tube containing 1000 μL of brain extract for either 1 s (group 1), 10 min (group 2), or 1 h (group 3). Additional bots collected from CWD non-detect white-tailed deer were incubated overnight in a $10^{-3}$ dilution of a brain extract from a prion-free tg1536 mouse. After completion of the incubation times, the bots were placed over a paper towel and air-dried. Then, bots were homogenized and tested for the presence of seeding competent prions by PMCA. All experiments were performed in three biological replicates.

## CWD-prions adsorption to nasal bots

Serial brain dilutions ($10^{-3}$, $10^{-5}$, and $10^{-8}$ brain dilutions) were prepared from a clinically sick CWD-infected tg1536 homozygous mouse brain (master homogenate prepared at 10% w/v as mentioned above). These brain dilutions were used on nasal bots collected from three different CWD non-detect white-tailed deer. Prior to the experiments, bots were washed with PBS to eliminate traces of blood and mucus. The prion adsorption procedure was performed by placing one bot in a tube containing 1000 μL of brain extract overnight under gentle rotation. As controls, additional bots collected from CWD non-detect white-tailed deer were incubated in a $10^{-3}$ brain dilution from a prion-free tg1536 homozygous mouse. After completing the process bots were washed and air-dried before storing them at −20 °C in new Eppendorf tubes. Bots

were homogenized and serially diluted before being tested by PMCA as mentioned above. All experiments were performed in three biological replicates.

## CWD prion exposure to grass plants

Wheatgrass plants were seeded in 10 g of commercially available compost soil (uncle Jim's worm farm) spiked with 833 µL of 20% (w/v) bot homogenates. Parasites were collected from CWD-infected and CWD non-detect deer. Wheatgrass seeds were kept for 4 days in the dark (covered with aluminum foil) to promote germination. Then, the seed-containing receptacles were transferred into a red-blue light indoor system and exposed to 12-h light/dark cycle for 4 weeks. Plants were kept at 23 °C at all times. The compost was constantly wet with tap water and the watering regime was maintained constant for all groups. Experimental and control plants were grown in separate compartments to avoid cross-contamination. After 4 weeks, plants were collected and leaves and roots were carefully separated using disposable tweezers and blades for each plant specimen. Roots were washed three to four times with PBS at room temperature to eliminate compost particles. After washing, the roots were dried by placing them in a soft and absorbent paper wipe. Roots and leaves were individually placed in 1.5-mL microcentrifuge tubes and stored at −20 °C until use.

## Evaluation of CWD prion content in plants

Plant tissues were homogenized at 10% (w/v) with extraction buffer containing a Roche® complete EDTA-free protease inhibitor cocktail. Samples were pre-treated with phosphotungstic sodium acid (NaPTA) at 0.57% (v/v) final concentration and incubated at 37 °C overnight with shaking (600 rpm in an Eppendorf® thermomixer). Then, samples were centrifuged for 30 min at $15,000 \times g$ and 10 °C. The pellets were washed with 200 µL of a solution containing 0.1% w/v sarkosyl and 500 mM EDTA and centrifuged for 10 min at $16,000 \times g$ and 10 °C. Pellets were stored at −20 °C until use. For measurements, pellets were resuspended in 90 µL of PMCA substrate and submitted to PMCA as explained above.

## Evaluation of CWD prion content in soils

Five soil samples of 0.1 g each were taken separately and homogenized at 20% w/v in PBS containing a Roche® Complete EDTA-free protease inhibitor cocktail. These samples were placed in an orbital shaker overnight. Then, homogenization was performed in a Precellys® 24 homogenizer using Hard Tissue Homogenizer tubes. This process was carried out for both the control and experimental groups. Finally, 10 µL of resuspended soil extracts were mixed with 90 µL of PMCA substrate and submitted to PMCA as explained above.

# Data availability

This study includes no data deposited in external repositories.

# Peer review information

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

## Acknowledgements

This work was supported by grants from the NIH (R01AI132695) and the USDA (AP20VSSPRS00C143) to RM. The authors would like to thank Dr. Edward Hoover (Colorado State University) for providing a piece of brain from an experimentally CWD-infected, terminally ill white/tailed deer. We would also like to thank Dr. Glenn Telling (Colorado State University) for providing tg1536 breeders to initiate our colony, and Ms. Ana Maria Benavides Obon for her assistance in processing bot specimens for histological analyses.

## Author contributions

**Paulina Soto**: Data curation; Formal analysis; Investigation; Writing—review and editing. **Francisca Bravo-Risi**: Data curation; Investigation. **Carlos Kramm**: Investigation. **Nazaret Gamez**: Investigation. **Rebeca Benavente**: Investigation. **Denise L Bonilla**: Formal analysis; Writing—review and editing. **J Hunter Reed**: Resources; Writing—review and editing. **Mitch Lockwood**: Resources. **Terry R Spraker**: Data curation; Investigation. **Tracy Nichols**: Conceptualization; Resources; Writing—review and editing. **Rodrigo Morales**: Conceptualization; Supervision; Funding acquisition; Methodology; Writing—original draft; Project administration.

## Disclosure and competing interests statement

