## [Peer Review File · EMBO Reports]

Nasal bots carry relevant titers of CWD prions in naturally infected white-tailed deer.

Paulina Soto, Francisca Bravo-Risi, Carlos Kramm, Nazaret Gamez, Rebeca Benavente, Denise Bonilla, J Hunter Reed, Mitch Lockwood, Terry Spraker, Tracy Nichols, and Rodrigo Morales

DOI: [10.15252/embr.202357382](https://doi.org/10.15252/embr.202357382)

Corresponding author(s): *Rodrigo Morales (Rodrigo.MoralesLoyola@uth.tmc.edu)*

Review Timeline:

Submission Date:	24th Apr 23
Editorial Decision:	17th May 23
Revision Received:	1st Sep 23
Editorial Decision:	29th Sep 23
Revision Received:	31st Oct 23
Accepted:	7th Nov 23

Editor: *Ioannis Papaioannou / Achim Breiling*

Transaction Report:

Dear Rodrigo,

Thank you for the transfer of your research manuscript for consideration by EMBO reports. It has now been seen by four experts in the field, and we have received the full set of their reports that are included below.

As you will see, all referees are positive about the manuscript, and they acknowledge that the findings are well-presented and interesting, and that most experiments are well-performed and appropriately controlled. However, they also identify some limitations that need to be addressed in a revised version of the manuscript. Notably, they all raise concerns regarding the conclusion that nasal bots concentrate infectious prions. They argue that this claim is not substantiated, and it should therefore either be supported by more conclusive experimental data or toned down, according to the advice of referee #4. Furthermore, the reviewers provide a number of suggestions for the improvement of the study and the manuscript (including the addition of some controls and important references, clarification of some parts, and correction of typos and punctuation mistakes).

Given these constructive comments, we would like to invite you to revise your manuscript with the understanding that the referee concerns (as detailed above and in their reports) must be fully addressed and their suggestions taken on board. Please address all referee concerns in a complete point-by-point response. Acceptance of the manuscript will depend on a positive outcome of a second round of review. It is EMBO reports policy to allow a single round of revision only and acceptance or rejection of the manuscript will therefore depend on the completeness of your responses included in the next, final version of the manuscript. If you have any questions or comments, we can also discuss the revisions in a video chat, if you like.

We realize that it is difficult to revise to a specific deadline. In the interest of protecting the conceptual advance provided by the work, we usually recommend a revision within 3 months (August 16th). Please discuss with me the revision progress ahead of this time if you require more time to complete the revisions.

IMPORTANT NOTE:

We perform an initial quality control of all revised manuscripts before re-review. Your manuscript will FAIL this control and the handling will be DELAYED if the following APPLIES:

- 1) If a data availability section providing access to data deposited in public databases is missing. If you have not deposited any data, please add a sentence to the data availability section that explains that (see below for more information).
- 2) If your manuscript contains statistics and error bars based on $n=2$. Please use scatter plots in these cases. No statistics should be calculated if $n=2$.

- 1) A .docx formatted version of the manuscript text (including legends for main figures, EV figures and tables). Please make sure that the changes are highlighted to be clearly visible.
- 2) Individual production quality figure files as .eps, .tif, .jpg (one file per figure). Please download our Figure Preparation Guidelines (figure preparation pdf) from our Author Guidelines pages <https://www.embopress.org/page/journal/14693178/authorguide> for more info on how to prepare your figures.
- 3) A .docx formatted letter INCLUDING the reviewers' reports and your detailed point-by-point responses to their comments. As part of the EMBO Press transparent editorial process, the point-by-point response is part of the Review Process File (RPF), which will be published alongside your paper unless you opt out of this (please see below for further information).
- 4) A complete author checklist, which you can download from our author guidelines (<<https://www.embopress.org/page/journal/14693178/authorguide>>). Please insert information in the checklist that is also reflected in the manuscript. The completed author checklist will also be part of the RPF (please see below for more information).
- 5) Please note that all corresponding authors are required to supply an ORCID ID for their name upon submission of a revised manuscript (<<https://orcid.org/>>). Please find instructions on how to link your ORCID ID to your account in our manuscript tracking system in our Author guidelines (<<https://www.embopress.org/page/journal/14693178/authorguide#authorshipguidelines>>)

6) We replaced Supplementary Information with Expanded View (EV) Figures and Tables that are collapsible/expandable online. A maximum of 5 EV Figures can be typeset. EV Figures should be cited as 'Figure EV1, Figure EV2' etc... in the text and their respective legends should be included in the main text after the legends of regular figures.

- For the figures that you do NOT wish to display as Expanded View figures, they should be bundled together with their legends in a single PDF file called *Appendix*, which should start with a short Table of Content (including page numbers). Appendix figures should be referred to in the main text as: "Appendix Figure S1, Appendix Figure S2" etc. See detailed instructions regarding expanded view here:

<<https://www.embopress.org/page/journal/14693178/authorguide#expandedview>>

7) Please note that a "Data availability" section at the end of Materials and Methods is mandatory. In case you have no data that require deposition in a public database, please state so instead of refereeing to the database: "Our study includes no data deposited in public repositories." under the heading "Data availability".

See also <<https://www.embopress.org/page/journal/14693178/authorguide#dataavailability>>. Please note that the Data availability statement is restricted to new primary data that are part of this study.

8) We request authors to consider both actual and perceived competing interests. Please review the new policy (<<https://www.embopress.org/competing-interests>>) and update your competing interests statement if necessary. Please name this section 'Disclosure and competing interests statement' and place it after the Acknowledgements section.

9) Figure legends and data quantification:

- the name of the statistical test used to generate error bars and P values,
- the number (n) of independent experiments (please specify technical or biological replicates) underlying each data point,
- the nature of the bars and error bars (s.d., s.e.m.)
- If the data are obtained from n {less than or equal to} 2, use scatter plots showing the individual data points.

Discussion of statistical methodology can be reported in the Materials and Methods section, but figure legends should contain a basic description of n, P and the test applied.

10) We now request publication of original source data with the aim of making primary data more accessible and transparent to the reader. Our source data coordinator will contact you to discuss which figure panels we would need source data for and will also provide you with helpful tips on how to upload and organize the files.

11) Our journal encourages inclusion of *data citations in the reference list* to directly cite datasets that were re-used and obtained from public databases. Data citations in the article text are distinct from normal bibliographical citations and should directly link to the database records from which the data can be accessed. In the main text, data citations are formatted as follows: "Data ref: Smith et al, 2001" or "Data ref: NCBI Sequence Read Archive PRJNA342805, 2017". In the Reference list, data citations must be labeled with "[DATASET]". A data reference must provide the database name, accession number/identifiers and a resolvable link to the landing page from which the data can be accessed at the end of the reference. Further instructions are available at <<https://www.embopress.org/page/journal/14693178/authorguide#referencesformat>>.

12) Please also note our reference format:

<<http://www.embopress.org/page/journal/14693178/authorguide#referencesformat>>.

13) We now use CRediT to specify the contributions of each author in the journal submission system. CRediT replaces the author contribution section, which should be removed from the manuscript. Please use the free text box to provide more detailed descriptions. See also guide to authors:

<<https://www.embopress.org/page/journal/14693178/authorguide#authorshipguidelines>>.

14) As part of the EMBO publications' Transparent Editorial Process, EMBO reports publishes online a Review Process File to accompany accepted manuscripts. This File will be published in conjunction with your paper and will include the referee reports, your point-by-point response and all pertinent correspondence relating to the manuscript.

You can opt out of this by letting the editorial office know (emboreports@embo.org). If you do opt out, the Review Process File link will point to the following statement: "No Review Process File is available with this article, as the authors have chosen not to make the review process public in this case."

I look forward to seeing a revised version of your manuscript when it is ready. Please let me know if you have any questions or comments regarding the revision.

Best regards,

Ioannis

Ioannis Papaioannou, PhD
Editor
EMBO reports

Referee #1:

In this paper, Soto et. al. attempt to demonstrate that nasal bots carry sufficient titers of chronic wasting disease prions to transmit disease in nature. The manuscript for the most part is well written but should be edited for minor reference and punctuation errors. In short, the preliminary results lend support that nasal bots contain prion titers adequate to transmit disease. Two subjects need further attention: 1.) the role of nasal epithelium/secretions and 2.) the section on nasal bots concentrating prions.

Major Points:

1. Within the first paragraph of the results section, it is stated that the use of nasal epithelium is used to confirm presence of CWD prions. Could the authors provide insight into where/which nasal epithelium (respiratory/olfactory) was used for these assays? Was this consistent across all animals? In conjunction, the authors state in the second paragraph of 'nasal bots contain prion seeding activity' that "these parasites mostly feed from the secretions released by these tissues". Are the authors familiar with the Kraft et. al. JGV paper (Jan 2023) that discusses prion shedding in nasal secretions? This seems particularly relevant here. Can the authors demonstrate a correlation between the retropharyngeal lymph node and obex results and the nasal bot results? Can the authors then further comment on the differences between the bioassay of terminal CWD brain and the nasal bots (Figure 3A/Table 2)?
2. The conclusions drawn in the section on nasal bots concentrating infectious prions appears to be misaligned with the results presented. The data demonstrates that nasal bots incubated in CWD-positive extracts bind quantities of prions that are recoverable by 2 rounds of PMCA but there is no control PMCA to reference too. Could the authors please provide an explanation as to why the used tg1536 mouse brain for these experiments instead of the CWD(+) white-tailed deer brain? Additionally, could the authors provide a serial dilution western blot on the brain extract used to incubate the bots in for reference? Perhaps the word concentrate is incorrect for what is trying to be conveyed here?

Minor points:

1. References: Some additions to the third introduction paragraph may be helpful in supporting this body of work. The authors may want to include their new paper on 'Ticks harbor and excrete CWD' and perhaps update the blood references (Soto 17', 20', McNulty 20'). Could the authors please add a date to the Colwell reference in the section of CWD-contaminated bots with soils and plants?
2. First introduction paragraph: 4 Canadian provinces have CWD, not 3.
3. First paragraph of results: Please add in 'Figure 1B' for reference.
4. Could the authors please clarify Table 1 column stating 'CWD on site' where MW-1 and Del Rio both state 'No' but the text in results states that CWD is present but in low prevalence (3% and low - respectively).
5. Please correct sentence structure in the first sentence of interaction of CWD-Contaminated bots with soils and plants.
6. Figure 5 B & C: This figure is slightly confusing as the blots in B are aligned negative and positive results, but C is aligned positive then negative results.

Referee #2:

Soto et al. present convincing evidence to demonstrate nasal bot fly larva collected from CWD-infected deer have both prion infectivity (as measured by mouse bioassay) and prion seeding-activity (measured by PMCA). This is the first study to confirm bot flies can carry prion infectivity. The presence of prion infectivity in bot pupa provides an additional source of environmental CWD prions and a potential mechanism for direct CWD transmission via consumption of pupa or resulting flies harboring CWD prions. In future studies, it will be important to determine if first instar larvae from infected bot flies carry sufficient prion infectivity to directly transmit CWD to naïve hosts, as this direct instillation of first instar larvae into the nose and mouth of cervids could be a very efficient means of CWD transmission if prion titers are sufficient.

The study is very interesting, and most experiments are well performed and include the necessary controls. I do have several areas that could use clarification and/or revision.

Unfortunately, the manuscript does not have line numbers, so drawing attention to specific areas of concern will be somewhat difficult.

Concerns described in order of appearance.

Page 4, Introduction, end of paragraph 2. "importantly neglected" should be revised to "understudied" or a statement more clear. Parasites have not been entirely neglected in prion transmission, as the authors themselves describe in the following paragraph.

Results, page 12, Under the Nasal bots concentrate infectious prions section- The authors state that the parasites have higher infectivity than nasal mucosa based on Fig 1. Figure 1 shows only brain and nasal epithelium data, no bot data is presented in Fig 1 to allow this comparison. Figure 2 does show both bot data and nasal epithelium, but in these figures the NE reacts more rapidly during PMCA than bot material, undermining their statement. The text and conclusions regarding NE vs bot levels of infectivity should be revised. To truly comment on this comparison a bioassay of the NE tissue should have also been performed.

Discussion, page 15, paragraph 2. The text "It is important to mention that tg1536+/- mice overexpress the white-tailed deer PrPC, so PrPSc quantities in the brains of these mice are considerable higher than those found in naturally infected cervids" should be referenced or the data shown. In many models where PrP is overexpressed the opposite outcome occurs, where PrPSc at end stage disease is reduced compared to models with physiologic expression. (tga20 and tg7 mice for example)

Discussion, page 16, bottom paragraph. Has there ever been a documented observation of deer ingesting whole bots or shells? If so, please add a reference to strengthen this postulation. It is also very unlikely that roots will be consumed in most grazing situations, although again not impossible.

Methods, page 21, bioassay section. There is a sentence describing post-op recovery from stereotactic inoculation linked directly to monitoring for prion clinical signs at a much later time. These ideas should not be described in the same sentence. Text regarding monitoring for prion associated signs could be more specific than "periodically".

Figure 1 legend- third line. "Infectious particles" should be replaced with "PrPSc"

Figure 3C legend and Figure 3C do not correlate in the numbering schemes for C1-C20.

Figure 6. It is difficult to appreciate that direct ingestion of pupa, pupa cases or adult flies can occur based on the current drawing.

Figure S2. A higher magnification inset would be nice for the pharyngeal pouch panel. The current figure looks more like a stain precipitate artifact rather than tissue specific IHC. It would also be nice to know how many sections of bot fly GI tract were analyzed and the ratio of positive sections.

Figure S5. Typos occur in the row headers for cerebellum and hypothalamus.

Referee #3:

This manuscript by Soto et al describes analyses of bot fly larvae from CWD-infected and uninfected deer for the presence of prions. This novel work addresses the possibility that bot flies and their larvae can mediate transmission of CWD. Understanding the modes of CWD transmission is important towards the development of strategies to limit CWD's uncontrolled spread among

cervids. The current results provide convincing evidence that bot larvae can pick up CWD prions from infected hosts at levels that might mediate inter-cervid transmissions. For the most part the presentation is clear and effective, and the findings highly relevant to the prion field. However, I have a few concerns/suggestions for improving some aspects of this study and presentation.

1) p5: "...first instar larvae are ejected within a liquid into the host while the adult is in flight." This description could benefit from clarification. I have trouble visualizing how the liquid is ejected into the host while the fly is flying. Is it flying in the nasal cavity?

2) p9: Appendix Figure S2: This figure needs to show staining of negative controls, too, because the "positive" staining shown looks somewhat like artifacts that are sometimes seen in IHC.

3) The authors repeatedly argue that nasal bots concentrate prions, but their data supporting this was somewhat indirect and weak. It seems to me that in the experiment described on p12 and in Figure 4, a more proper comparison to reveal the tendency of bots to concentrate prions would be to compare bots transiently dipped into the CWD brain extract (allowing little time for anything other than wetting of the bot surface) to bots left overnight in the extract (during which time they could take up, or adsorb, prions). Otherwise, it is difficult to know the difference between what was simply carried over in the extract that wetted the surface of the bots from that actively adsorbed or ingested by the bots over time.

4) p13, Figure 5B: I'm not finding methods for how the soil samples were handled or extracted for PMCA testing.

Referee #4:

The work reported is carefully done and includes appropriate controls.

This work is noteworthy in that it demonstrates a potential role for parasites in the transmission and persistence of infectious prion material in the environment. It is worth noting, however, that this is not a new method of transmission or spread of infectious material in the environment (the same deer that are shedding bots are also shedding infectious material through other excreta).

Revisions:

1: The claim that bots concentrate PrPSc is not substantiated. If the bots did indeed concentrate PrPSc, then it would be necessary to demonstrate the concentration of PrPSc be higher in the bot than in the surrounding environment. This was not demonstrated (eg. when bots were incubated with known-positive brain homogenate).

A more appropriate descriptor would be accumulate. I understand if there is resistance to use the term as in this field we use accumulate to describe what happens to PrPSc that has been converted from native PrPC in a given animal. However...the experiments in this report do NOT demonstrate that bots concentrate PrPSc, but they DO demonstrate the bots accumulate PrPSc when exposed. This needs to be appropriately communicated.

2: The authors also contend that bots 'enhance infectivity'. This was not empirically demonstrated and that statement should be removed.

3: This report uses the terminology 'pre-clinical' as well as 'non-clinical'. The term non-clinical is appropriate, the term pre-clinical is not appropriate as these samples came from animals that were euthanized without clinical signs of disease.

Medical School
Department of Neurology

Houston, August 31st 2023

Dear Editor,

Please find enclosed the revised version of our manuscript entitled "**Nasal bots carry relevant titers of Chronic Wasting Disease (CWD) prions in naturally infected white-tailed deer.**". We are very glad that reviewers liked the article and we made several modifications to it following their suggestions. We believe that their constructive comments helped to make this article much stronger.

Next, we include a point-by-point answer to the reviewer's comments:

REFEREE #1:

Major Points

1. Within the first paragraph of the results section, it is stated that the use of nasal epithelium is used to confirm presence of CWD prions. Could the authors provide insight into where/which nasal epithelium (respiratory/olfactory) was used for these assays? Was this consistent across all animals? In conjunction, the authors state in the second paragraph of 'nasal bots contain prion seeding activity' that "these parasites mostly feed from the secretions released by these tissues". Are the authors familiar with the Kraft *et al.* JGV paper (Jan 2023) that discusses prion shedding in nasal secretions? This seems particularly relevant here. Can the authors demonstrate a correlation between the retropharyngeal lymph node and obex results and the nasal bot results? Can the authors then further comment on the differences between the bioassay of terminal CWD brain and the nasal bots (Figure 3A/Table 2)?

Answer: We appreciate the Reviewer's comment and are grateful for these suggestions.

- a. The nasal epithelium used in these assays was mostly composed by respiratory structures, as it was not collected at anatomically areas that are enriched in olfactory epithelium. The seeding activity depicted in the manuscript corresponded to a single deer. A second tissue was tested as shown in Figure 2, with similar results. Unfortunately, and due to the opportunistic nature of tissue collection, no more nasal epithelium specimens were analyzed in this study. This information has been added in the revised version of this manuscript.
- b. We thank the reviewer for pointing the work of Kraft *et al.* as it is timely and relevant for this particular study. In this revised version, we discuss several of our result considering the findings communicated in that article.
- c. Unfortunately, we were unable to correlate data between the test results on retropharyngeal lymph node and obex with PMCA results on nasal bots. This was due to the lack of information on these parameters for most of the donor deer. In future studies, we plan to address the effect of deer genotype and progression of the incubation periods

in the contamination of these parasites. This is particularly relevant considering the recently published data by Kraft *et al.* This limitation is now noted in the manuscript.

- d. In regards to the bioassay, we used a brain extract prepared from a clinically sick, CWD infected white-tailed deer. Tg1536 mice (expressing the deer PrP) intracerebrally infected with this material showed prion disease averaging 313 days. The administration of a nasal bot shell's homogenate also induced disease in all challenged mice but with extended incubation periods. According to our estimations, this represents approximately 10-100 infectivity logs of difference between both samples. Unfortunately, we were unable to obtain brain specimens from bot-donor deer. This refrained us to compare bot infectivity with other tissues in the same animal. We agree that this is an excellent question that we were unable to address in the current study. We have noted this limitation in the Discussion section of the revised article.

2. The conclusions drawn in the section on nasal bots concentrating infectious prions appears to be misaligned with the results presented. The data demonstrates that nasal bots incubated in CWD-positive extracts bind quantities of prions that are recoverable by 2 rounds of PMCA but there is no control PMCA to reference too. Could the authors please provide an explanation as to why the used tg1536 mouse brain for these experiments instead of the CWD(+) white-tailed deer brain? Additionally, could the authors provide a serial dilution western blot on the brain extract used to incubate the bots in for reference? Perhaps the word concentrate is incorrect for what is trying to be conveyed here?.

Answer: The response to this question is provided in different sections. These are itemized below:

- a. Although Figure 1B (brain serial dilutions) can be used as a reference for this experiment, it is relevant to know that we changed the format of this figure considering the questions from several Reviewers. We hope that you are satisfied with the replies provided (please refer to Question 3 from Reviewer #3 for reference).
- b. Brains from CWD-infected tg1536 mice were used considering their availability. Brains from CWD deer are scarce in our laboratory. Considering the availability of tg1536-CWD prions in our laboratory, we performed most of the *in vitro* analyses using this material. Previous reports and our own unpublished data demonstrate that prions from cervidized mice retain the infectious properties of the parental source. Considering this, we are confident that the results obtained in our *in vitro* studies using tg1536 mice's brains are translatable to deer-derived infectious particles. Some of the above statements have been added to the revised version of the manuscript.
- c. A western blot image comparing the levels of CWD prions in tg1536 mice and white-tailed deer is now provided as Supplementary Figure.
- d. In regards to the use of the word "concentrate", we agree with the reviewer. We have changed this word accordingly, and revised the text and interpretations. We have also added a new figure that analyze, in a more comprehensive manner, the interaction of nasal bots with prions. Please see the question 3 from Reviewer #3 for additional reference.

Minor points:

1. References: Some additions to the third introduction paragraph may be helpful in supporting this body of work. The authors may want to include their new paper on 'Ticks

harbor and excrete CWD' and perhaps update the blood references (Soto 17', 20', McNulty 20'). Could the authors please add a date to the Colwell reference in the section of CWD-contaminated bots with soils and plants?

Answer: We appreciate the Reviewer's comment and suggestion and we are grateful for the input. We added our last paper as a reference to support this section, and updated/edited the other references.

2. First introduction paragraph: 4 Canadian provinces have CWD, not 3.

Answer: Thanks for this one. We have updated this information now.

3. First paragraph of results: Please add in 'Figure 1B' for reference.

Answer: We have revised the first paragraph as indicated by the Reviewer and we added "Figure 1B" in the corresponding section.

4. Could the authors please clarify Table 1 column stating 'CWD on site' where MW-1 and Del Rio both state 'No' but the text in results states that CWD is present but in low prevalence (3% and low - respectively).

Answer: We apologize for this mistake. Although all the donor animals from Del Rio were diagnosed as CWD non-detect, surveillance data has identified a low prevalence of infected animals in the area. A similar situation was experienced at the MW-1 site (3% prevalence but all bot donor animals were CWD non-detect). This mistake has been corrected in the new version of this manuscript by stating "high prevalence" or "low prevalence" in the table. Additional adjustments were made in the text.

5. Please correct sentence structure in the first sentence of interaction of CWD-Contaminated bots with soils and plants.

Answer: This has been corrected as suggested.

6. Figure 5 B & C: This figure is slightly confusing as the blots in B are aligned negative and positive results, but C is aligned positive then negative results.

Answer: Figures 5 B and C have been revised and modified for clarity.

REFEREE #2:

Concerns described in order of appearance.

1. Page 4, Introduction, end of paragraph 2. "importantly neglected" should be revised to "understudied" or a statement more clear. Parasites have not been entirely neglected in prion transmission, as the authors themselves describe in the following paragraph.

Answer: We have changed this statement as suggested by the Reviewer.

2. Results, page 12, Under the Nasal bots concentrate infectious prions section- The authors state that the parasites have higher infectivity than nasal mucosa based on Fig 1. Figure 1 shows only brain and nasal epithelium data, no bot data is presented in Fig 1 to allow this comparison. Figure 2 does show both bot data and nasal epithelium, but in these figures the NE reacts more rapidly during PMCA than bot material, undermining their statement. The text and conclusions regarding NE vs bot levels of infectivity should be revised. To truly comment on this comparison a bioassay of the NE tissue should have also been performed.

Answer: We understand the Reviewer concern and completely agree with it. According to our experience, we expect that both nasal mucosa and bot extracts will not have the same efficiency on PMCA as the brain extracts considering that their matrices may alter the performance of the assay. In addition, taking into account the ultrasensitive nature of PMCA, CWD prion quantifications in these different samples are not adequate. The Reviewer's point is well taken and we have modified the text, accordingly. Specifically, we have removed the statements regarding the incongruency between nasal epithelium and nasal bot infectivity. We are really grateful to this Reviewer for pointing this issue.

3. Discussion, page 15, paragraph 2. The text "It is important to mention that tg1536+/- mice overexpress the white-tailed deer PrPC, so PrPSc quantities in the brains of these mice are considerable higher than those found in naturally infected cervids" should be referenced or the data shown. In many models where PrP is overexpressed the opposite outcome occurs, where PrPSc at end stage disease is reduced compared to models with physiologic expression. (tga20 and tg7 mice for example)

Answer: We now provide additional information comparing the PrP^{Sc} levels in brains from the terminally ill deer and tg1536 mice used in this study. Specifically, this was performed via western blots in brains extracts subjected to proteinase K digestion. As shown in Supplementary Figure 8, this analysis shows that terminally ill tg1536 mice have approximately 10 times more PrP^{Sc} compared with their deer counterpart (this analysis is restricted to the materials used in this study). This information is now incorporated in the revised version of this manuscript.

4. Discussion, page 16, bottom paragraph. Has there ever been a documented observation of deer ingesting whole bots or shells? If so, please add a reference to strengthen this postulation. It is also very unlikely that roots will be consumed in most grazing situations, although again not impossible.

Answer: There are no documented observations of deer ingesting bots or shells directly from the ground, although it is well acknowledged that in time they are pulverized and incorporated in the environment. As mentioned by the Reviewer, roots are not commonly consumed by deer, but their ingestion increase in winter when normally available food is scarce. We understand that the statements in the previous version need to be treated with caution. For that reason, we have rephrased them for accuracy.

5. Methods, page 21, bioassay section. There is a sentence describing post-op recovery from stereotactic inoculation linked directly to monitoring for prion clinical signs at a much later time. These ideas should not be described in the same sentence. Text regarding monitoring for prion associated signs could be more specific than "periodically".

Answer: This specific text has been modified to note the specificity of the monitoring schedule, as suggested.

6. Figure 1 legend- third line. "Infectious particles" should be replaced with "PrPSc"

Answer: We modified the text as recommended by the Reviewer.

7. Figure 3C legend and Figure 3C do not correlate in the numbering schemes for C1-C20.

Answer: We thank the reviewer for catching this. We have fixed this mistake accordingly.

8. Figure 6. It is difficult to appreciate that direct ingestion of pupa, pupa cases or adult flies can occur based on the current drawing.

Answer: For clarity, we have added a new Supplementary Figure describing these events. We have also modified the text to better describe the possibility of these events to occur.

9. Figure S2. A higher magnification inset would be nice for the pharyngeal pouch panel. The current figure looks more like a stain precipitate artifact rather than tissue specific IHC. It would also be nice to know how many sections of bot fly GI tract were analyzed and the ratio of positive sections.

Answer: We understand the concerns. We added a new Supplementary Figure that include increased magnifications of prion deposits in the pharyngeal pouches. As appreciated, these panels depict the typical deposition of disease-associated prion protein in tissues. For bots, we analyzed five tissue sections per bot. All sections of parasites collected from CWD positive white-tailed deer analyzed by IHC displayed at least one PrP^{Sc} positive deposit. However, we acknowledge that the number of parasites analyzed by this method was limited and future studies should also focus on IHC tests.

10. Figure S5. Typos occur in the row headers for cerebellum and hypothalamus.

Answer: We have fixed this mistake, accordingly.

REFEREE #3:

1. p5: "...first instar larvae are ejected within a liquid into the host while the adult is in flight." This description could benefit from clarification. I have trouble visualizing how the liquid is ejected into the host while the fly is flying. Is it flying in the nasal cavity?

Answer: We apologize for not making this section clear and we revised the article for clarity. We also added a reference where this process is better explained. We also added a Supplementary Figure explaining this process. In summary, this process starts by fertilized female flies being attracted to deer by means of their scent or exhaled CO₂. Female flies deposit packages of 30 to 50 eggs (L1) in the nostrils of deer. The eggs are surrounded by a thick, gelatinous liquid that promotes their adhesion and protects them from drying out. The endogenous life cycle begins when the eggs (L1 diapause) move into the nose or mouth towards the nasal cavities aided by their hooks and spines which constitute a defense mechanism against the host's attempts to get rid of them through coughs, sneezes, and sudden movements of the head. It is relevant to note

that L1s can enter in a phase of hypobiosis or diapause as an adaptive response to adverse weather conditions. As the parasites mature, they progressively travel to the retropharyngeal pouches and they are finally expelled from deer within a liquid to continue their maturation in soils. This information has not been added to the manuscript, but it is provided to the Reviewer to clarify this specific concern.

2) p9: Appendix Figure S2: This figure needs to show staining of negative controls, too, because the "positive" staining shown looks somewhat like artifacts that are sometimes seen in IHC.

Answer: Please refer to Question 9 from Referee #2.

3) The authors repeatedly argue that nasal bots concentrate prions, but their data supporting this was somewhat indirect and weak. It seems to me that in the experiment described on p12 and in Figure 4, a more proper comparison to reveal the tendency of bots to concentrate prions would be to compare bots transiently dipped into the CWD brain extract (allowing little time for anything other than wetting of the bot surface) to bots left overnight in the extract (during which time they could take up, or adsorb, prions). Otherwise, it is difficult to know the difference between what was simply carried over in the extract that wetted the surface of the bots from that actively adsorbed or ingested by the bots over time.

Answer: We appreciate the question and we agree with the concern. To address this concern, new experiments were performed and were added as main figures of this manuscript. In the first study, bots collected from CWD non-detect deer were transiently exposed to a CWD-laden brain extract at different times ranging from 1 second to 1 hour. After 1 PMCA round, we retrieved PrP^{Sc} signals in all the time points, although the group of bots exposed for the longest time showed a higher frequency of positivity in the different replicates tested. In a second PMCA round, all bots from all groups displayed positive signals. This data suggests that bots can bind prions even after a transient exposure to CWD prions. In addition, we performed a second experiment in which bots collected from CWD non-detect deer were exposed overnight to different concentration of prions. To better appreciate the ability of bots to interact with CWD prions, the exposed parasites were homogenized and serially diluted before being tested in the PMCA assay. In this experiment, we observed that bots exposed to the 10^{-3} and 10^{-5} CWD-laden brain extracts displayed positive signals even after bot extracts were diluted 10^6 times. Interestingly, bots exposed to a 10^{-8} CWD brain extract were detected only in its lower dilution (10^{-2}). It is relevant to note that a 10^{-8} CWD brain extract approaches to the range of the limiting detection by PMCA, suggesting that nasal bots are able to bind little quantities of prions. This information has now been added in the revised version of this manuscript.

4) p13, Figure 5B: I'm not finding methods for how the soil samples were handled or extracted for PMCA testing.

Answer: We apologize for this oversight. We now provide a detailed explanation of the protocol used for this experiment. This can be found in Materials and Methods.

REFEREE #4

1. The claim that bots concentrate PrPSc is not substantiated. If the bots did indeed concentrate PrPSc, then it would be necessary to demonstrate the concentration of PrPSc be higher in the bot than in the surrounding environment. This was not demonstrated (eg. when bots were incubated with known-positive brain homogenate). A more appropriate descriptor would be accumulate. I understand if there is resistance to use the term as in this field we use accumulate to describe what happens to PrPSc that has been converted from native PrPC in a given animal. However...the experiments in this report do NOT demonstrate that bots concentrate PrPSc, but they DO demonstrate the bots accumulate PrPSc when exposed. This needs to be appropriately communicated.

Answer: We completely agree with the Reviewer and appreciate the input. We have modified the text, as suggested. Please refer to Question 3 from Reviewer #3 for further details.

2. The authors also contend that bots 'enhance infectivity'. This was not empirically demonstrated and that statement should be removed.

Answer: We have removed this statement as suggested.

3. This report uses the terminology 'pre-clinical' as well as 'non-clinical'. The term non-clinical is appropriate, the term pre-clinical is not appropriate as these samples came from animals that were euthanized without clinical signs of disease.

Answer: Again, we agree. Thanks for the input and clarification. We have modified, accordingly.

We are really grateful for the useful comments. In summary, we carefully considered the reviewers' suggestions and addressed their comments, we improved several figures, modified the text, performed additional experiments and added 4 new figures that further support our conclusions. We believe that all suggestions and comments have made this study much stronger.

I look forward to hear from you.

Rodrigo Morales, PhD

Professor

Department of Neurology

The University of Texas Health Science Center at Houston

Dear Dr. Morales,

Thank you for the submission of your revised manuscript to our editorial offices. I have now received the reports from the three referees that have been asked to re-evaluate your paper, you will find below. As you will see, the referees now fully support the publication of the study in EMBO reports. Referees #1 and #3 have some remaining concerns and suggestions to improve the study, I ask you to address in a final revised manuscript.

- Please provide a final title with not more than 100 characters (including spaces)
- Please provide the abstract written in present tense throughout.
- Please add up to five keywords to the title page, below the abstract.
- We now use CRediT to specify the contributions of each author in the journal submission system. CRediT replaces the author contribution section. Please use the free text box to provide more detailed descriptions and do NOT provide your final manuscript text file with an author contributions section. See also our guide to authors: <https://www.embopress.org/page/journal/14693178/authorguide#authorshippinguidelines>
- We updated our journal's competing interests policy in January 2022 and request authors to consider both actual and perceived competing interests. Please review the policy <https://www.embopress.org/competing-interests> and update your competing interests if necessary. Please name this section 'Disclosure and Competing Interests Statement' and put it after the Acknowledgements section.
- Per journal policy, we do not allow 'data not shown', which is stated in the manuscript. All data referred to in the paper should be displayed in the main or Expanded View figures, or an Appendix. Thus, please add these data (or change the text accordingly if these data are not central to the study). See: <http://embor.embopress.org/authorguide#unpublisheddata>
- Please provide individual production quality figure files as .eps, .tif, .jpg (one file per figure), of the main figures. Please upload these as separate, individual files upon re-submission.

Please consult our guide for figure preparation (and follow the size and format requirements):

- Please add scale bars of similar style and thickness to the microscopic images (main and Appendix figures), using clearly visible black or white bars (depending on the background). Please place these in the lower right corner of the images themselves. Please do not write on or near the bars in the image but define the size in the respective figure legend. Presently, some panels lack scale bars (e.g. S3) or contain text (S2).
- Please correct the scale bar unit in panel 3C from μM to μm .
- Please format the figure legends according to our journal style. See the respective section in our guide to authors (please find the link below). Please separate each panel description by a line brake and make sure that the panels are listed in alphabetic order. Moreover, please add to each legend a 'Data Information' section explaining the statistics used or providing information regarding replicates and scale.

- Please make sure that all figure panels are called out separately and sequentially. Presently, there seems to be no separate callout for Figure 2A and B and 5D and E. Please check.
- As Figure 2 has only one panel, please remove the label A.
- There is no legend for panel 5E. Please check.

- I would suggest moving the two tables to the Appendix file. Please name these Appendix Table S1 and Appendix Table S2, provide a legend and add the tables to the Appendix table of contents. Finally, please update the callouts for these tables in the main manuscript text file.

- Regarding data quantification and statistics, please make sure that the number "n" for how many independent experiments were performed, their nature (biological versus technical replicates), the bars and error bars (e.g. SEM, SD) and the test used to calculate p-values is indicated in the respective figure legends (main and Appendix figures). Please also check that all the p-values are explained in the legend, and that these fit to those shown in the figure. Please provide statistical testing where applicable. Please avoid the phrase 'independent experiment', but clearly state if these were biological or technical replicates. Please also indicate (e.g. with n.s.) if testing was performed, but the differences are not significant. In case n=2 please show the data as separate datapoints or bars without error bars and statistics. See also:
<http://www.embopress.org/page/journal/14693178/authorguide#statisticalanalysis>

- Please use our reference format:
<http://www.embopress.org/page/journal/14693178/authorguide#referencesformat>

In addition, I would need from you:

Best,

Referee #1:

Thank you to the authors for responding to all previous requests. The answers were sufficient to accept the manuscript with minor points listed below.

Minor points:

1. Line 54: I believe there are 31 states now positive.
2. Line 117: C. phobifer is listed twice, once in parentheses.
3. Line 122: Sentence starting with unlike appears to have a word missing.
4. Line 306/307: Both sentences begin with 'As expected'. Perhaps change one?
5. Line 324: Cavity is missing the 'y'

Referee #2:

All concerns have been well addressed

Referee #3:

With respect to the authors' response to my specific concerns:

1) Thanks for the explanation.

2) Comparable stains of negative control tissue is still needed.

3) The new experiments improve the quantitative understanding of prion uptake by the larvae. However, the authors' conclusion in L318-320 "Overall, these data 319 suggest that nasal bots are able to accumulate prion infectivity, regardless of the quantity of 320 prions to which they are exposed" would be less ambiguous if "from their environment" were added after 'infectivity'.

Otherwise the reader might interpret the statement as meaning that the larvae can themselves replicate, rather than simply bind, prions.

All editorial and formatting issues were resolved by the authors.

Dr. Rodrigo Morales
The University of Texas Health Science Center at Houston
Neurology
6431 Fannin St.
MSB 7.218
Houston, TX 77030
United States

Dear Dr. Morales,

I am very pleased to accept your manuscript for publication in the next available issue of EMBO reports. Thank you for your contribution to our journal.

Yours sincerely,
